# LARP: Tokenizing Videos with a Learned Autoregressive Generative Prior

**Hanyu Wang,  Saksham Suri,  Yixuan Ren,  Hao Chen**[*,†]**,  Abhinav Shrivastava**[*]
University of Maryland, College Park
{hywang66, sakshams, yxren}@umd.edu  {chenh, abhinav}@cs.umd.edu
Project page: https://hywang66.github.io/larp/

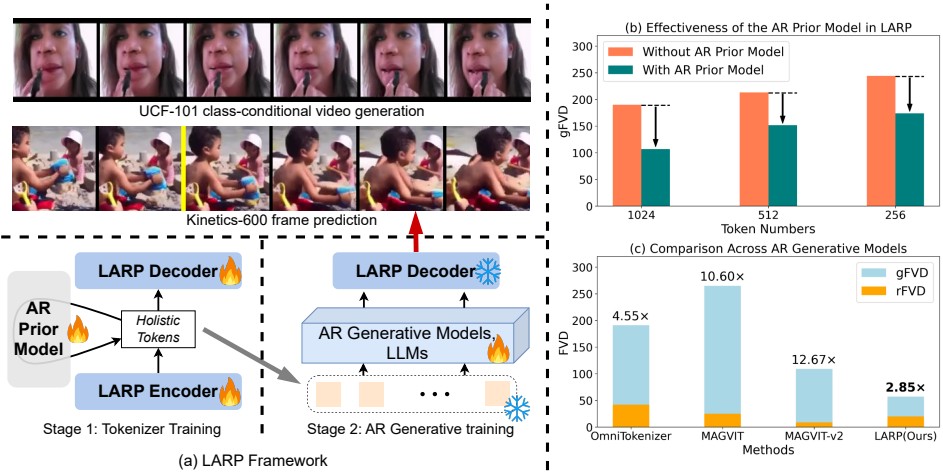

Figure 1:    **LARP highlights.** **(a)** LARP is a video tokenizer for two-stage video generative models. In the first stage, LARP tokenizer is trained with a lightweight AR prior model to learn an AR-friendly latent space. In the second stage, an AR generative model is trained on LARP's discrete tokens to synthesize high-fidelity videos. **(b)** The incorporation of the AR prior model significantly improves the generation FVD (gFVD) across various token number configurations. **(c)** LARP shows a much smaller gap between its reconstruction FVD (rFVD) and generation FVD (gFVD), indicating the effectiveness of the optimized latent space it has learned.

## Abstract

We present LARP, a novel video tokenizer designed to overcome limitations in current video tokenization methods for autoregressive (AR) generative models. Unlike traditional patchwise tokenizers that directly encode local visual patches into discrete tokens, LARP introduces a holistic tokenization scheme that gathers information from the visual content using a set of learned holistic queries. This design allows LARP to capture more global and semantic representations, rather than being limited to local patch-level information. Furthermore, it offers flexibility by supporting an arbitrary number of discrete tokens, enabling adaptive and efficient tokenization based on the specific requirements of the task. To align the discrete token space with downstream AR generation tasks, LARP integrates a lightweight AR transformer as a training-time prior model that predicts the next token on its discrete latent space. By incorporating the prior model during training, LARP learns a latent space that is not only optimized for video reconstruction but is also structured in a way that is more conducive to autoregressive generation. Moreover, this process defines a sequential order for the discrete tokens, progressively pushing them toward an optimal configuration during training, ensuring smoother and more accurate AR generation at inference time. Comprehensive experiments demonstrate LARP's strong performance, achieving state-of-the-art FVD on the UCF101 class-conditional video generation benchmark. LARP enhances the compatibility of AR models with videos and opens up the potential to build unified high-fidelity multimodal large language models (MLLMs).

---

[*]Co-corresponding authors. † Now a research scientist at ByteDance.

# 1 INTRODUCTION

The field of generative modeling has experienced significant advancements, largely driven by the success of autoregressive (AR) models in the development of large language models (LLMs) (Bai et al., 2023; Brown, 2020; Radford et al., 2019; Google et al., 2023; Touvron et al., 2023a;b). Building on AR transformers (Vaswani, 2017), these models are considered pivotal for the future of AI due to their exceptional performance (Hendrycks et al., 2020; 2021), impressive scalability (Henighan et al., 2020; Kaplan et al., 2020; Rae et al., 2021), and versatile flexibility (Radford et al., 2019; Brown, 2020).

Inspired by the success of LLMs, recent works have begun to employ AR transformers for visual generation (Van Den Oord et al., 2017; Razavi et al., 2019; Esser et al., 2021; Hong et al., 2022; Ge et al., 2022; Kondratyuk et al., 2023; Wang et al., 2024). Additionally, several recent developments have extended LLMs to handle multimodal inputs and outputs (Lu et al., 2022; Zheng et al., 2024), further demonstrating the promising potential of AR models in visual content generation. All of these methods employ a **visual tokenizer** to convert continuous visual signals into sequences of discrete tokens, allowing them to be autoregressively modeled in the same way as natural language is modeled by LLMs. Typically, a visual tokenizer consists of a visual encoder, a quantization module (Van Den Oord et al., 2017; Yu et al., 2023b), and a visual decoder. The generative modeling occurs in the quantized discrete latent space, with the decoder mapping the generated discrete token sequences back to continuous visual signals. It is evident that the visual tokenizer plays a pivotal role, as it directly influences the quality of the generated content. Building on this insight, several works have focused on improving the visual tokenizer (Lee et al., 2022; Yu et al., 2023b), making solid progress in enhancing the compression ratio and reconstruction fidelity of visual tokenization.

Most existing visual tokenizers follow a patchwise tokenization paradigm (Van Den Oord et al., 2017; Esser et al., 2021; Wang et al., 2024; Yu et al., 2023b), where the discrete tokens are quantized from the encoded patches of the original visual inputs. While these approaches are intuitive for visual data with spatial or spatialtemporal structures, they restrict the tokenizers' ability to capture global and holistic representations of the entire input. This limitation becomes even more pronounced when applied to AR models, which rely on sequential processing and require locally encoded tokens to be transformed into linear 1D sequences. Previous research (Esser et al., 2021) has demonstrated that the method of flattening these patch tokens into a sequence is critical to the generation quality of AR models. Although most existing works adopt a raster scan order for this transformation due to its simplicity, it remains uncertain whether this is the optimal strategy. In addition, there are no clear guidelines for determining the most effective flattening order.

On the other hand, although the reconstruction fidelity of a visual tokenizer sets an upper bound on the generation fidelity of AR models, the factors that determine the gap between them remain unclear. In fact, higher reconstruction quality has been widely reported to sometimes lead to worse generation fidelity (Zhang et al., 2023; Yu et al., 2024). This discrepancy highlights the limitations of the commonly used reconstruction-focused design of visual tokenizers and underscores the importance of ensuring desirable properties in the latent space of the tokenizer. However, very few works have attempted to address this aspect in improving image tokenizers (Gu et al., 2024; Zhang et al., 2023), and for video tokenizers, it has been almost entirely overlooked.

In this paper, we present LARP, a video tokenizer with a **L**earned **A**uto**R**egressive generative **P**rior, designed to address the underexplored challenges identified in previous work. By leveraging a ViT-style spatialtemporal patchifier (Dosovitskiy, 2020) and a transformer encoder architecture (Vaswani, 2017), LARP forms an autoencoder and employs a stochastic vector quantizer (Van Den Oord et al., 2017) to tokenize videos into holistic token sequences. Unlike traditional patchwise tokenizers, which directly encode input patches into discrete tokens, LARP introduces a set of learned queries (Carion et al., 2020; Li et al., 2023) that are concatenated with the input patch sequences and then encoded into holistic discrete tokens. An illustrative comparison between the patchwise tokenizer and LARP is shown in Figure 2 (a) and the left part of Figure 2 (b). By decoupling the direct correspondence between discrete tokens and input patches, LARP allows for a flexible number of discrete tokens, enabling a trade-off between tokenization quality and latent representation length. This design also empowers LARP to produce more holistic and semantic representations of video content.

To further align LARP's latent space with AR generative models, we incorporate a lightweight AR transformer as a prior model. It autoregressively models LARP's latent space during training, providing signals to encourage learning a latent space that is well-suited for AR models. Importantly, the prior model is trained simultaneously with the main modules of LARP, but it is discarded during inference, adding zero memory or computational overhead to the tokenizer. Notably, by combining holistic tokenization with the co-training of the AR prior model, LARP automatically determines an order for latent discrete tokens in AR generation and optimizes the tokenizer to perform optimally within that structure. This approach eliminates the need to manually define a flattening order, which remains an unsolved challenge for traditional tokenizers.

To evaluate the effectiveness of the LARP tokenizer, we train a series of Llama-like (Touvron et al., 2023a;b; Sun et al., 2024) autoregressive (AR) generation models. Leveraging the holistic tokens and the learned AR generative prior, LARP achieves a Frechét Video Distance (FVD) (Unterthiner et al., 2018) score of 57 on the UCF101 class-conditional video generation benchmark (Soomro, 2012), establishing a new state-of-the-art among all published video generative models, including proprietary and closed-source approaches like MAGVIT-v2 (Yu et al., 2023b). To summarize, our key contributions are listed as follows:

- We present LARP, a novel video tokenizer that enables flexible, holistic tokenization, allowing for more semantic and global video representations.
- LARP features a learned AR generative prior, achieved by co-training an AR prior model, which effectively aligns LARP's latent space with the downstream AR generation task.
- LARP significantly improves video generation quality for AR models across varying token sequence lengths, achieving state-of-the-art FVD performance on the UCF101 class-conditional video generation benchmark and outperforming all AR methods on the K600 frame prediction benchmark.

## 2 RELATED WORK

### 2.1 DISCRETE VISUAL TOKENIZATION

To enable AR models to generative high resolution visual contents, various discrete visual tokenization methods have been developed. The seminal work VQ-VAE (Van Den Oord et al., 2017; Razavi et al., 2019) introduces vector quantization to encode continuous images into discrete tokens, allowing them to be modeled by PixelCNN (Van den Oord et al., 2016). VQGAN (Esser et al., 2021) improves visual compression rate and perceptual reconstruction quality by incorporating GAN loss (Goodfellow et al., 2014) in training the autoencoder. Building on this, several works focus on improving tokenizer efficiency (Cao et al., 2023) and enhancing generation quality (Gu et al., 2024; Zheng et al., 2022; Zhang et al., 2023). Leveraging the powerful ViT (Dosovitskiy, 2020) architecture, ViT-VQGAN (Yu et al., 2021) improves VQGAN on image generation tasks.

Inspired by the success of image tokenization, researchers extend VQGAN to videos using 3D CNNs (Ge et al., 2022; Yan et al., 2021; Yu et al., 2023a). C-ViViT (Villegas et al., 2022) employs the temporal-causal ViT architecture to tokenize videos, while more recent work, MAGVIT-v2 (Yu et al., 2023b), introduces lookup-free quantization, significantly expanding the size of the quantization codebook. OmniTokenizer (Wang et al., 2024) unifies image and video tokenization using the same tokenizer model and weights for both tasks.

It is worth noting that all of the above tokenizers follow the patchwise tokenization paradigm discussed in Section 1, and are therefore constrained by patch-to-token correspondence. Very recently, a concurrent work (Yu et al., 2024) proposes a compact tokenization approach for images. However, it neither defines a flattening order for the discrete tokens nor introduces any prior or regularization to improve downstream generation performance.

### 2.2 VISUAL GENERATION

Visual generation has been a long-standing area of interest in machine learning and computer vision research. The first major breakthrough comes with the rise of Generative Adversarial Networks (GANs) (Goodfellow et al., 2014; Karras et al., 2019; 2020; Skorokhodov et al., 2022), known for their intuitive mechanism and fast inference capabilities. AR methods are also widely applied in

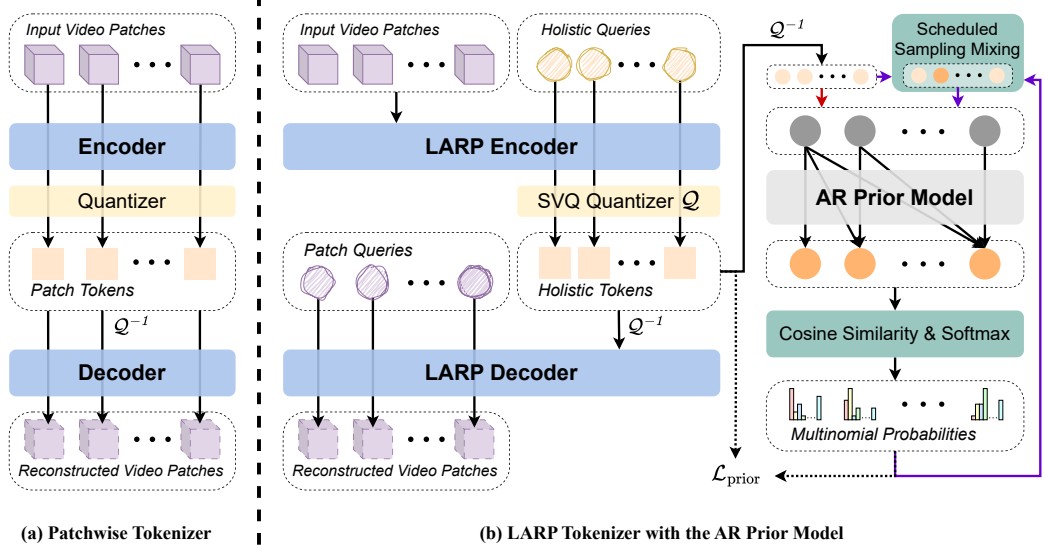

Figure 2: **Method overview.** Cubes ▱ represent video patches, circles ○ indicate continuous embeddings, and squares □ denote discrete tokens. **(a) Patchwise video tokenizer** used in previous works. **(b) Left: The LARP tokenizer** tokenizes videos in a holistic scheme, gathering information from the video using a set of learned queries. **Right: The AR prior model**, trained with LARP, predicts the next holistic token, enabling a latent space optimized for AR generation. The AR prior model is forwarded in two rounds per iteration. The **red** arrow represents the first round, and the **purple** arrows represent the second round. The reconstruction loss $\mathcal{L}_{\text{rec}}$ is omitted for simplicity.

visual generation. Early works (Van Den Oord et al., 2016; Van den Oord et al., 2016; Chen et al., 2020) model pixel sequences autoregressively, but are limited in their ability to synthesize high-resolution content due to the extreme length of pixel sequences. Recent advancements in visual tokenization make AR generative models for visual content more practical. While all tokenizers discussed in Section 2.1 are suitable for AR generation, many focus on BERT-style (Devlin, 2018) masked visual generation (Chang et al., 2022), such as in Yu et al. (2023a;b; 2024). Diffusion models (Ho et al., 2020; Song et al., 2020; Peebles & Xie, 2023) have recently emerged to dominate image (Dhariwal & Nichol, 2021) and video synthesis (Ho et al., 2022), delivering impressive visual generation quality. By utilizing VAEs (Kingma, 2013) to reduce resolution, latent diffusion models (Rombach et al., 2022; Blattmann et al., 2023) further scale up, enabling multimodal visual generation (Betker et al., 2023; Saharia et al., 2022; Podell et al., 2023; Brooks et al., 2024).

## 3 METHOD

### 3.1 PRELIMINARY

**Patchwise Video Tokenization.** As discussed in Section 1, existing video tokenizers adopt a patchwise tokenization scheme, where latent tokens are encoded from the spatialtemporal patches of the input video. Typically, a patchwise video tokenizer consists of an encoder $\mathcal{E}$, a decoder $\mathcal{D}$, and a quantizer $\mathcal{Q}$. Given a video input $\mathbf{V} \in \mathbb{R}^{T \times H \times W \times 3}$, it is encoded, quantized, and reconstructed as:

$$\mathbf{Z} = \mathcal{E}(\mathbf{V}), \qquad \mathbf{X} = \mathcal{Q}(\mathbf{Z}), \qquad \hat{\mathbf{V}} = \mathcal{D}(\mathbf{X}), \tag{1}$$

where $\mathbf{Z} \in \mathbb{R}^{\frac{T}{f_T} \times \frac{H}{f_H} \times \frac{W}{f_W} \times d}$ refers to the spatialtemporally downsampled video feature maps with $d$ latent dimensions per location, $\mathbf{X} \in \mathbb{N}^{T' \times H' \times W'}$ denotes the quantized discrete tokens, and $\hat{\mathbf{V}}$ is the reconstructed video. $f_T, f_H, f_W$ are the downsampling factors for the spatialtemporal dimensions $T, H, W$, respectively.

Despite different implementations of the encoder $\mathcal{E}$, decoder $\mathcal{D}$, and quantizer $\mathcal{Q}$, all patchwise tokenizers maintain a fixed downsampling factor for each spatialtemporal dimension. The latent vector $\mathbf{Z}_{i,j,k,:} \in \mathbb{R}^d$ at each position is typically the direct output of its spatialtemporally corresponding

input video patch (e.g., same spatialtemporal location in CNNs, or token position in transformers). While this design is intuitive for 3D signals like video, it limits the discrete tokens to low-level patch features, hindering their ability to capture higher-level, holistic information. Moreover, this formulation introduces the challenge of flattening patch tokens into a unidirectional sequence, which is critical for AR generation.

**Autoregressive Modeling.** Given a sequence of discrete tokens $\boldsymbol{x} = (x_1, x_2, \ldots, x_n)$, we can train a neural network to model the probability distribution $p_\theta(\boldsymbol{x})$ autoregressively as follows:

$$p_\theta(\boldsymbol{x}) = \prod_{i=1}^{n} p_\theta\left(x_i \mid x_1, \ldots, x_{i-1}, \theta\right), \tag{2}$$

where $\theta$ denotes the neural network parameters. This model can be conveniently trained by optimizing the negative log-likelihood (NLL) of $p_\theta(\boldsymbol{x})$. During inference, it iteratively predicts the next token $x_i$ by sampling from $p_\theta\left(x_i \mid x_1, \ldots, x_{i-1}, \theta\right)$, based on the previously generated tokens.

While autoregressive modeling imposes no direct constraints on data modality, it does require the data to be both *discrete* and *sequential*, which necessitates the use of a visual tokenizer when applied to images or videos.

## 3.2 Holistic Video Tokenization

**Patchify.** LARP employs the transformer architecture (Vaswani, 2017) due to its exceptional performance and scalability. Following the ViT framework (Dosovitskiy, 2020), we split the input video into spatialtemporal patches, and linearly encode each patch into continuous transformer patch embeddings. Formally, given a video input $\mathbf{V} \in \mathbb{R}^{T \times H \times W \times 3}$, the video is linearly patchified as follows:

$$\mathbf{P} = \mathcal{P}(\mathbf{V}), \quad \boldsymbol{E} = \text{flatten}(\mathbf{P}), \tag{3}$$

where $\mathcal{P}$ denotes the linear patchify operation, $\mathbf{P} \in \mathbb{R}^{\frac{T}{f_T} \times \frac{H}{f_H} \times \frac{W}{f_W} \times d}$ is the spatialtemporal patches projected onto $d$ dimensions, and $\boldsymbol{E} \in \mathbb{R}^{m \times d}$ is the flattened $d$-dimentional patch embeddings. Here, $f_T, f_H, f_W$ are the downsampling factors for dimensions $T, H, W$, respectively, and $m = \frac{T}{f_T} \times \frac{H}{f_H} \times \frac{W}{f_W}$ is the total number of tokens. Importantly, the patch embeddings $\boldsymbol{E}$ remain local in nature, and therefore cannot be directly used to generate holistic discrete tokens.

**Query-based Transformer.** To design a holistic video tokenizer, it is crucial to avoid directly encoding individual patches into discrete tokens. To achieve this, we adapt the philosophy of Carion et al. (2020); Li et al. (2023) to learn a set of fixed input queries to capture the holistic information from the video, as illustrated in the left section of Figure 2 (b). For simplicity, LARP employs a transformer encoder [1] architecture, as opposed to the transformer encoder-decoder structure used in Carion et al. (2020). In-context conditioning is applied to enable information mixing between different patch and query tokens.

Formally, we define $n$ learnable holistic query embedding $\boldsymbol{Q}_L \in \mathbb{R}^{n \times d}$, where each embedding is $d$-dimensional. These query embeddings are concatenated with the patch embeddings $\boldsymbol{E}$ along the token dimension. The resulting sequence, now of length $(n+m)$, is then input to the LARP encoder $\mathcal{E}$ and quantizer $\mathcal{Q}$ as follows:

$$\boldsymbol{Z} = \mathcal{E}(\boldsymbol{Q}_L \parallel \boldsymbol{E}), \quad \boldsymbol{x} = \mathcal{Q}(\boldsymbol{Z}_{1:n,:}), \tag{4}$$

where $\parallel$ denotes the concatenation operation, $\boldsymbol{Z}$ is the latent embeddings, and $\boldsymbol{x} = (x_1, \ldots, x_n)$ denotes the quantized discrete tokens. Note that only $\boldsymbol{Z}_{1:n,:}$, i.e., the latent embeddings corresponding to the queries embeddings, are quantized and used. This ensures that each discrete token $x_i$ has equal chance to represent any video patch, eliminating both soft and hard local patch constraints.

The LARP decoder is also implemented as a transformer encoder neural network. During the decoding stage, LARP follows a similar approach, utilizing $m$ learnable patch query embeddings $\boldsymbol{Q}_P \in \mathbb{R}^{m \times d}$. The decoding process is defined as:

$$\hat{\boldsymbol{Z}} = \mathcal{Q}^{-1}(\boldsymbol{x}), \quad \hat{V} = \text{reshape}(\mathcal{D}(\boldsymbol{Q}_P \parallel \hat{\boldsymbol{Z}})_{1:m,:}), \tag{5}$$

---

[1] Here and throughout this paper, "transformer encoder" refers to the specific parallel transformer encoder architecture defined in Dosovitskiy (2020)

where $\mathcal{Q}^{-1}$ denotes the de-quantization operation that maps discrete tokens $x$ back to the continuous latent embeddings $\hat{Z} \in \mathbb{R}^{n \times d}$. These embeddings are concatenated with the patch query embeddings $Q_P$, and the combined sequence of length $m + n$ is decoded into a sequence of continuous vectors. The first $m$ vectors are reshaped to reconstruct the video $\hat{V} \in \mathbb{R}^{T \times H \times W \times 3}$.

Crucially, although the latent tokens $x$ are now both holistic and discrete, no specific flattening order is imposed due to the unordered nature of the holistic query set and the parallel processing property of the transformer encoder. As a result, $x$ is not immediately suitable for AR modeling.

**Stochastic Vector Quantization.** While vector quantization (VQ) (Van Den Oord et al., 2017) has been widely adopted in previous visual quantizers (Esser et al., 2021; Ge et al., 2022), its deterministic nature limits the tokenizer's ability to explore inter-code correlations, resulting less semantically rich codes. To address these limitations, LARP employs a stochastic vector quantization (SVQ) paradigm to implement the quantizer $\mathcal{Q}$. Similar to VQ, SVQ maintains a codebook $C \in \mathbb{R}^{c \times d'}$, which stores $c$ vectors, each of dimension $d'$. The optimization objective $\mathcal{L}_{\text{SVQ}}$ includes a weighted sum of the commitment loss and the codebook loss, as defined in Van Den Oord et al. (2017). The key difference lies in the look-up operation. While VQ uses an $\arg\min$ operation to find the closest code by minimizing the distance between the input vector $v \in \mathbb{R}^{d'}$ and all codes in $C$, SVQ introduces stochasticity in this process. Specifically, SVQ computes the cosine similarities $s$ between the input vector $v$ and all code vectors in $C$, interprets these similarities as logits, and applies a softmax normalization to obtain the probabilities $p$. One index $x$ is then sampled from the resulting multinomial distribution $P(x)$. Formally, the SVQ process $x = \mathcal{Q}(v)$ is defined as:

$$s = \frac{v \cdot C_i}{\|v\|\|C_i\|}, \quad p = \text{softmax}(s), \tag{6}$$

$$x \sim P(x) = \prod_{j=1}^{n} p_i^{\mathbf{1}_{x=j}}, \tag{7}$$

where $\mathbf{1}$ denotes the indicator function. To maintain the differentiability of SVQ, we apply the straight-through estimator (Bengio et al., 2013). The de-quantization operation is performed via a straightforward index look-up, $\hat{v} = \mathcal{Q}^{-1}(x) = C_x$, similar to the standard VQ process.

**Reconstructive Training.** Following Esser et al. (2021); Ge et al. (2022); Yu et al. (2023a), the reconstructive training loss of LARP, $\mathcal{L}_{\text{rec}}$, is composed of $L_1$ reconstruction loss, LPIPS perceptual loss (Zhang et al., 2018), GAN loss (Goodfellow et al., 2014), and SVQ loss $\mathcal{L}_{\text{SVQ}}$.

### 3.3 LEARNING AN AUTOREGRESSIVE GENERATIVE PRIOR

**Continuous Autoregressive Transformer.** To better align LARP's latent space with AR generative models, we introduce a lightweight AR transformer as a prior model, which provides gradients to push the latent space toward a structure optimized for AR generation. A key challenge in designing the prior model lies in its discrete nature. Simply applying an AR model to the discrete token sequence would prevent gradients from being back-propagated to the LARP encoder. Furthermore, unlike the stable discrete latent spaces of fully trained tokenizers, LARP's latent space is continuously evolving during training, which can destabilize AR modeling and reduce the quality of the signals it provides to the encoder. To address these issues, we modify a standard AR transformer into a continuous AR transformer by redefining its input and output layers, as depicted in the right section of Figure 2 (b).

The input layer of a standard AR transformer is typically an embedding look-up layer. In the prior model of LARP, this is replaced with a linear projection that takes the de-quantized latents $\hat{Z}$ as input, ensuring proper gradient flow during training. The output layer of a standard AR transformer predicts the logits of the next token. While this does not block gradient propagation, it lacks awareness of the vector values in the codebook, making it unsuitable for the continuously evolving latent space during training. In contrast, the output layer of LARP's AR prior model makes predictions following the SVQ scheme described in Section 3.2. It predicts an estimate of the next token's embedding, $\bar{v} \in \mathbb{R}^{d'}$, which has the same shape as a codebook vectors $C_i$. Similar to SVQ, the predicted embedding $\bar{v}$ is used to compute cosine similarities with all code vectors in $C$, as described in Equation (6). These similarities are then softmax-normalized and interpreted as probabilities,

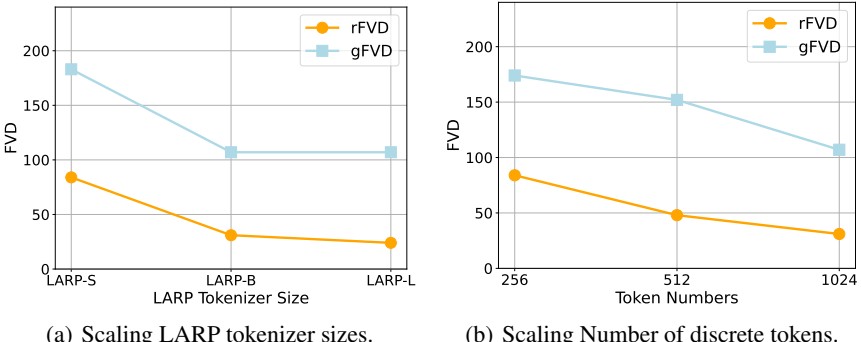

(a) Scaling LARP tokenizer sizes.    (b) Scaling Number of discrete tokens.

Figure 3: **Scaling LARP tokenizer size and number of tokens.**

which are used to compute the negative log-likelihood (NLL) loss with the input tokens as the ground truth. To predict the next token, a sample is drawn from the resulting multinomial distribution using Equation (7). This output layer design ensures that the AR prior model remains aware of the continuously evolving codebook, enabling it to make more accurate predictions and provide more precise signals to effectively train the LARP tokenizer.

**Scheduled Sampling.** Exposure bias (Ranzato et al., 2015) is a well-known challenge in AR modeling. During training, the model is fed the ground-truth data to predict the next token. However, during inference, the model must rely on its own previous predictions, which may contain errors, creating a mismatch between training and inference conditions. While the AR prior model in LARP is only used during training, it encounters a similar issue: as the codebook evolves, the semantic meaning of discrete tokens can shift, making the input sequence misaligned with the prior model's learned representations. To address this problem, we employ the scheduled sampling technique (Bengio et al., 2015; Mihaylova & Martins, 2019) within the AR prior model of LARP. Specifically, after the first forward pass of the prior model, we randomly mix the predicted output sequence with the original input sequence at the token level. This mixed sequence is then fed into the AR prior model for a second forward pass. The NLL loss is computed for both rounds of predictions and averaged, helping to reduce exposure bias and ensure more robust training.

**Integration.** Although the AR prior model functions as a standalone module, it is trained jointly with the LARP tokenizer in an end-to-end manner. Once the NLL loss $\mathcal{L}_{\text{prior}}$ is computed, it is combined with the reconstructive loss $\mathcal{L}_{\text{rec}}$ to optimize the parameters of both the prior model and the tokenizer. Formally, the total loss is defined as:

$$\mathcal{L} = \mathcal{L}_{\text{rec}} + \alpha \mathcal{L}_{\text{prior}}, \tag{8}$$

where $\alpha$ is the loss weight, and $\mathcal{L}_{\text{rec}}$ is defined in Section 3.2. Since $\alpha$ is is typically set to a small value, we apply a higher learning rate to the parameters of the prior model to ensure effective learning. Importantly, the prior model is used solely to encourage an AR-friendly discrete latent space for LARP during training. It is discarded at inference time, meaning it has no effect on the inference speed or memory footprint.

## 4 EXPERIMENTS

### 4.1 SETUP

**Dataset.** We conduct video reconstruction and generation experiments using the Kinetics-600 (K600)(Carreira et al., 2018) and UCF-101(Soomro, 2012) datasets. In all experiments, we use 16-frame video clips with a spatial resolution of $128 \times 128$ for both training and evaluation following Ge et al. (2022); Yu et al. (2023a;b).

**Implementation Details.** LARP first patchifies the input video. In all experiments, the patch sizes are set to $f_T = 4$, $f_H = 8$, and $f_W = 8$, respectively. As a result, a $16 \times 128 \times 128$ video clip is split into $4 \times 16 \times 16 = 1024$ video patches, which are projected into 1024 continuous patch embeddings in the first layer of LARP. For the SVQ quantizer, we utilize a factorized codebook with a size of

| Method | #Params | | #Tokens | rFVD↓ | gFVD↓ | |
|---|---|---|---|---|---|---|
| | Tokenizer | Generator | | | K600 | UCF |
| *Diffusion-based generative models with continuous video tokenizers* | | | | | | |
| VideoFusion (Luo et al., 2023) | - | 2B | - | - | - | 173 |
| HPDM (Skorokhodov et al., 2024) | - | 725M | - | - | - | 66 |
| *MLM generative models with discrete video tokenizers* | | | | | | |
| MAGVIT-MLM (Yu et al., 2023a) | 158M | 306M | 1024 | 25 | 9.9 | 76 |
| MAGVIT-v2-MLM (Yu et al., 2023b) | - | 307M | 1280 | **8.6** | **4.3** | 58 |
| *AR generative models with discrete video tokenizers* | | | | | | |
| CogVideo (Hong et al., 2022) | - | 9.4B | 2065 | - | 109.2 | 626 |
| TATS (Ge et al., 2022) | 32M | 321M | 1024 | 162 | - | 332 |
| MAGVIT-AR (Yu et al., 2023a) | 158M | 306M | 1024 | 25 | - | 265 |
| MAGVIT-v2-AR (Yu et al., 2023b) | - | 840M | 1280 | **8.6** | - | 109 |
| OmniTokenizer (Wang et al., 2024) | 82.2M | 650M | 1280 | 42 | 32.9 | 191 |
| LARP-L (Ours) | 173M | 343M | 1024 | 24 | 6.2 | 107 |
| LARP-L-Long (Ours) | 173M | 343M | 1024 | 20 | 6.2 | 102 |
| LARP-L-Long (Ours) | 173M | 632M | 1024 | 20 | 5.1 | **57** |

Table 1: **Comparison of video generation results.** Results are grouped by the type of generative models. The scores for MAGVIT-AR and MAGVIT-v2-AR are taken from the appendix of MAGVIT-v2 (Yu et al., 2023b). LARP-L-Long denotes the LARP-L trained for more epochs. Our best results are obtained with a larger AR generator.

8192 and a dimension of $d' = 8$, following the recommendations of Yu et al. (2021). The softmax normalization in Equation (6) is applied with a temperature of 0.03. The AR prior model in LARP is adapted from a small GPT-2 model (Radford et al., 2019), consisting of only 21.7M parameters. Scheduled sampling for the AR prior model employs a linear warm-up for the mixing rate, starting from 0 and reaching a peak of 0.5 at 30% of the total training steps. We set AR prior loss weight $\alpha = 0.06$ in our main experiments, and use a learning rate multiplier of 50.

We employ a Llama-like Touvron et al. (2023a;b); Sun et al. (2024) transformer as our AR generative model. One class token [cls] and one separator token [sep] are used in the class-conditional generation task on UCF101 and frame prediction task on K600, respectively.

Frechét Video Distance (FVD) (Unterthiner et al., 2018) serves as the main evaluation metric for both reconstruction and generation experiments.

## 4.2 SCALING

To explore the effect of scaling the LARP tokenizer, we begin by varying its size while keeping the number of latent tokens fixed at 1024. As shown in Figure 3 (a), we compare the reconstruction FVD (rFVD) and generation FVD (gFVD) for three scaled versions of LARP : LARP-L, LARP-B, and LARP-S, with parameter counts of 173.0M, 116.3M, and 39.8M, respectively. All results are reported on the UCF-101 dataset. Interestingly, while rFVD consistently improves as the tokenizer size increases, gFVD saturates when scaling from LARP-B to LARP-L, suggesting that gFVD can follow a different trend from rFVD. Notably, as shown in Figure 1 (c), LARP has already achieved the smallest gap between rFVD and gFVD, further demonstrating the effectiveness of the optimized latent space it has learned.

One of LARP's key features is its holistic video tokenization, which supports an arbitrary number of latent discrete tokens. Intuitively, using more tokens slows down the AR generation process but improves reconstruction quality. Conversely, using fewer tokens significantly speeds up the process but may lead to lower reconstruction quality due to the smaller information bottleneck. To evaluate this trade-off, we use LARP-B and the default AR model, scaling down the number of latent tokens from 1024 to 512 and 256. The corresponding rFVD and gFVD results on the UCF-101 dataset are reported in Figure 3 (b). It is expected that both rFVD and gFVD increase when fewer tokens are used to represent a video. However, the rate of degradation in gFVD slows down when reducing from 512 to 256 tokens compared to rFVD, indicating improved generative representation efficiency.

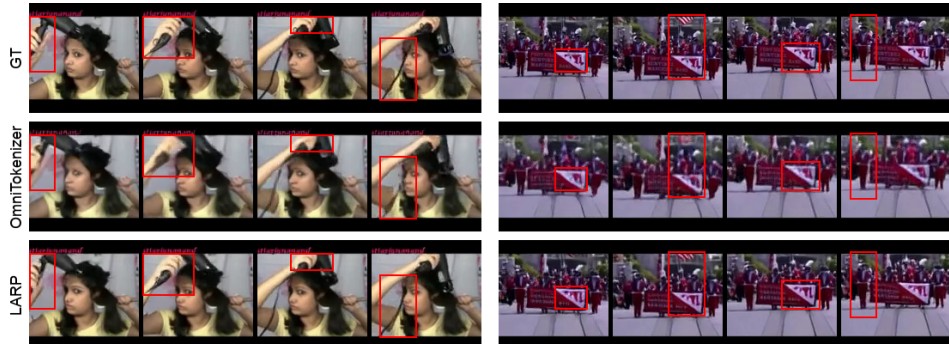

Figure 4: **Video reconstruction comparison** with OmniTokenizer (Wang et al., 2024).

### 4.3 VIDEO GENERATION COMPARISON

For video generation, we compare LARP with other state-of-the-art published video generative models, including diffusion-based models, Masked Language Modeling (MLM) methods, and AR methods. We use the UCF-101 class-conditional generation benchmark and the K600 frame prediction benchmark, where the first 5 frames are provided to predict the next 11 frames in a 16-frame video clip. As shown in Table 1, LARP outperforms all other video generators on the UCF-101 dataset, setting a new state-of-the-art FVD of 57. Notably, within the family of AR generative models, LARP significantly surpasses all other AR methods by a large margin on both the UCF-101 and K600 datasets, including the closed-source MAGVIT-v2-AR (Yu et al., 2023b). Moreover, the last two rows of Table 1 demonstrate that using a larger AR generator can significantly improve LARP's generation quality, highlighting the scalability of LARP's representation.

### 4.4 LATENT SPACE ANALYSIS

We analyze LARP's latent space by examining the role of individual LARP tokens in video reconstruction. Our findings show that only certain subsets of tokens have a significant impact on quality. The specific set of important tokens differs across videos, reflecting the content adaptive nature of LARP tokens as different subsets become active based on the video content. Additionally, spatial-temporal analysis reveals that LARP tokens function as holistic video representations, influencing global structures rather than isolated patches, with their effects aligning with semantic features rather than appearing random. Further details and visualizations are provided in Appendix B.3.

### 4.5 VISUALIZATION

**Video Reconstruction.** In Figure 4, we compare video reconstruction quality of LARP with OmniTokenizer (Wang et al., 2024). LARP consistently outperforms OmniTokenizer, particularly in complex scenes and regions, further validating the rFVD comparison results shown in Table 1.

**Class-Conditional Video Generation.** We present class-conditional video generation results in Figure 5. LARP constructs a discrete latent space that better suited for AR generation, which enables the synthesis of high-fidelity videos, not only improving the quality of individual frames but also enhancing overall temporal consistency. Additional results are provided in the appendix.

| Configuration | PSNR↑ | LPIPS↓ | rFVD↓ | gFVD↓ |
|---|---|---|---|---|
| LARP-B | 27.88 | 0.0855 | 31 | **107** |
| No AR prior model | **27.95** | **0.0830** | **23** | 190 |
| No scheduled sampling in AR prior model | 27.85 | 0.0856 | 27 | 142 |
| Deterministic quantization | 27.65 | 0.0884 | 27 | 149 |
| Small AR prior model loss weight ($\alpha = 0.03$) | 27.83 | 0.0866 | 28 | 120 |
| No CFG | 27.88 | 0.0855 | 31 | 121 |

Table 2: **Ablation study.** All configurations are modified from LARP-B model.

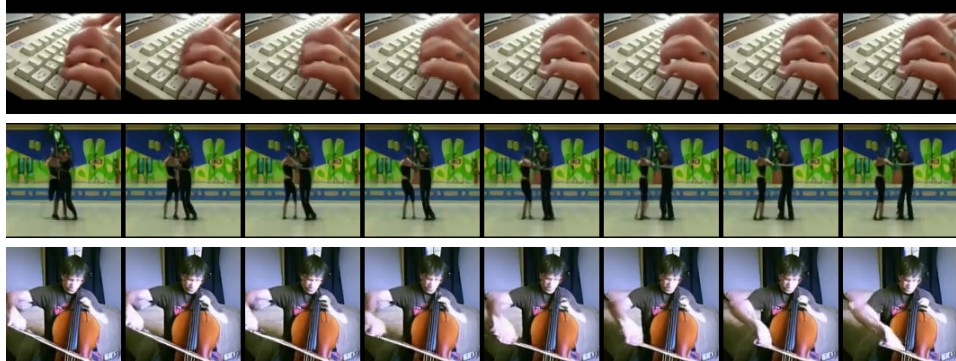

Figure 5: **Class-conditional video generation results** on the UCF-101 dataset using LARP.

**Video Frame Prediction.** Video frame prediction results are displayed in Figure 6. The vertical yellow line marks the boundary between the conditioned frames and the predicted frames. We use 5 frames as input to predict the following 11 frames, forming a 16-frame video clip, which is temporally downsampled to 8 frames for display. Additional results are provided in the appendix.

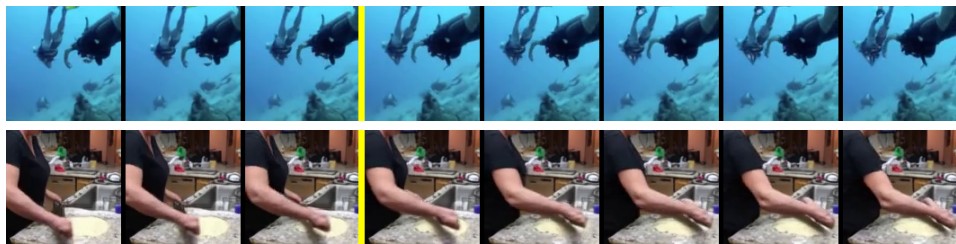

Figure 6: **Video frame prediction results** on the K600 dataset using LARP.

### 4.6 ABLATION STUDY

To assess the impact of the different components proposed in Section 3, we perform an ablation study, with results shown in Table 2. Clearly, the AR prior model contributes the most to the exceptional performance of LARP. As further validated in Figure 1 (b), the improvement from using the AR prior model remains consistent across different token numbers. The scheduled sampling for the AR prior model and the use of SVQ are also critical, as both are closely tied to the AR prior model's effectiveness. The loss weight of the AR prior model and the use of CFG have relatively minor effects on the generative performance. Interestingly, the model without the AR prior achieves the best reconstruction results but the worst generation results, highlighting the effectiveness of the AR prior model in *enhancing LARP's discrete latent space for generative tasks*.

## 5 CONCLUSION AND FUTURE WORK

In this paper, we introduce LARP, a novel video tokenizer tailored specifically for autoregressive (AR) generative models. By introducing a holistic tokenization scheme with learned queries, LARP captures more global and semantic video representations, offering greater flexibility in the number of discrete tokens. The integration of a lightweight AR prior model during training optimizes the latent space for AR generation and defines an optimal token order, significantly improving performance in AR tasks. Extensive experiments on video reconstruction, class-conditional video generation, and video frame prediction demonstrate LARP's ability to achieve state-of-the-art FVD scores. The promising results of LARP not only highlight its efficacy in video generation tasks but also suggest its potential for broader applications, including the development of multimodal large language models (MLLMs) to handle video generation and understanding in a unified framework.

ACKNOWLEDGMENTS

This work was partially supported by NSF CAREER Award (#2238769) and an Amazon Research Award (Fall 2023) to AS. The authors acknowledge UMD's supercomputing resources made available for conducting this research. The U.S. Government is authorized to reproduce and distribute reprints for Governmental purposes notwithstanding any copyright annotation thereon. The views and conclusions contained herein are those of the authors and should not be interpreted as necessarily representing the official policies or endorsements, either expressed or implied, of NSF, Amazon, or the U.S. Government.

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

# A ADDITIONAL IMPLEMENTATION DETAILS

## A.1 ADDITIONAL IMPLEMENTATION DETAILS OF THE LARP TOKENIZER.

During the training of LARP, a GAN loss (Goodfellow et al., 2014) is employed to enhance reconstruction quality. We use a ViT-based discriminator (Dosovitskiy, 2020) with identical patchify settings to those of the LARP tokenizer. The discriminator is updated once for every five updates of the LARP tokenizer and is trained with a learning rate that is 30% of the LARP tokenizer's learning rate. To stabilize discriminator training, LeCam regularization (Tseng et al., 2021) is applied, following the approach of Yu et al. (2023a). A GAN loss weight of 0.3 is used throughout the training.

Fixed sin-cos positional encoding (Vaswani, 2017) is used in both the encoder and decoder of LARP. In the encoder, fixed 3D positional encoding is applied to each video patch, while in the decoder, fixed 1D positional encoding is added to each holistic token. Notably, since the patch queries and holistic queries are position-wise learnable parameters, they do not require additional positional encodings.

In the SVQ module, we set the total quantization loss weight to 0.1. Additionally, we follow Esser et al. (2021) by using a commitment loss weight of 0.25 and a codebook loss weight of 1.0. Both the $L_1$ reconstruction loss and the LPIPS perceptual loss are assigned a weight of 1.0.

In most experiments, we train the LARP tokenizer for 75 epochs on a combined dataset of UCF-101 and K600 with a batch size of 64, totaling approximately 500k training steps. Random horizontal flipping is used as a data augmentation technique. Specifically, LARP-L-Long in Table 1 is trained for 150 epochs with a batch size of 128.

The Adam optimizer (Kingma, 2014) is used with a base learning rate of $1e-4$, $\beta_1 = 0.9$, and $\beta_2 = 0.95$, following a warm-up cosine learning rate schedule.

## A.2 ADDITIONAL IMPLEMENTATION DETAILS OF THE AR GENERATIVE MODEL

We use Llama-like transformers as our AR generative models. Unlike the original implementation and Sun et al. (2024), we utilize absolute learned positional encodings. A token dropout probability of 0.1 is applied during training, with both residual and feedforward dropout probabilities also set to 0.1. Additionally, when training the AR generative models, the SVQ module of the LARP tokenizer is set to be deterministic, ensuring a more accurate latent representation.

Our default AR generative model consists of 632M parameters, as specified in Table 1. It is trained on the training split of the UCF-101 dataset for 1000 epochs with a batch size of 32. The model used in the last row of Table 1, which also has 632M parameters, is trained for 3000 epochs on UCF-101 with a batch size of 64.

The AdamW optimizer (Loshchilov, 2017) is used with $\beta_1 = 0.9$, $\beta_2 = 0.95$, a weight decay of 0.05, and a base learning rate of $6e-4$, following a warm-up cosine learning rate schedule.

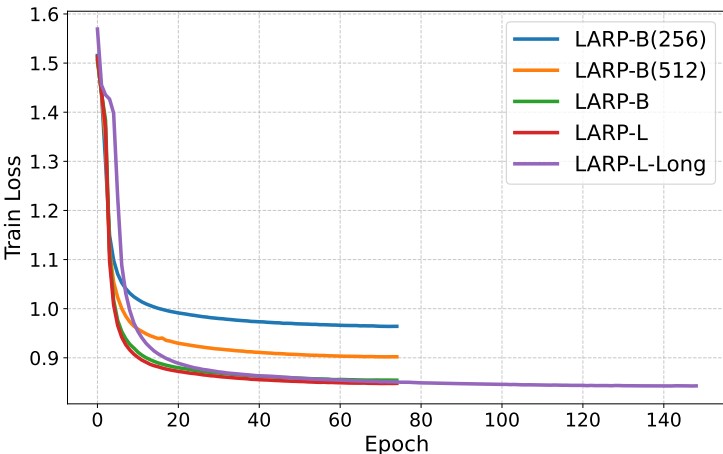

Figure 7: **Training loss of LARP under different configurations.**

When generating videos, we apply a small Classifier-Free Guidance (CFG) scale of 1.25 (Ho & Salimans, 2022). We do not use top-k or top-p sampling methods.

For the frame prediction task, to create the 5-frame conditioning holistic tokens, we first take the initial 5 frames from the ground truth video, then repeat the 5th frame 11 times and pad these repetitions with the initial 5 frames (using the same padding). This forms a complete 16-frame conditioning video. This conditioning video is then input to the LARP tokenizer to generate the conditioning holistic tokens. At inference time, the AR model predicts the holistic tokens for the full video, conditioned on the holistic tokens from the padded conditioning video.

## B ADDITIONAL EXPERIMENTS

### B.1 TRAINING STABILITY

While training LARP involves multiple loss functions, the process remains stable across different configurations. In Figure 7, we show the training loss curves for LARP under various training epochs and regimes. Here, LARP-B(256) and LARP-B(512) denote LARP -B trained with 256 and 512 holistic tokens, respectively. LARP-L-Long is trained with a batch size of 128 for 150 epochs, while the other models are trained with a batch size of 64 for 75 epochs.

As shown, all loss curves decrease smoothly and converge at the end of training. Note that the initial slope change in LARP-L-Long's curve is due to the linear warm-up learning rate schedule applied during the first 8 epochs.

### B.2 HOLISTIC REPRESENTATION

The holistic tokenization scheme of LARP enables global and semantic representations, effectively capturing the spatiotemporal redundancy in videos and representing them accurately within a compact latent space. To validate LARP's capability to represent videos globally and holistically, we manually introduce information redundancy into videos and measure the improvement in reconstruction quality.

Specifically, we fix the total number of video frames to 16, consistent with our main experiments, and repeat shorter videos multiple times to form 16-frame videos. This increases information redundancy as the number of unique frames decreases with higher repetition, implying improved reconstruction quality for tokenizers capable of exploiting global and holistic information. We also compare LARP against a patchwise video tokenizer, OmniTokenizer. Notably, although OmniTokenizer operates in a patchwise manner, its transformer architecture allows it to leverage some global information. For repetition levels, a value of 1 indicates no repetition, where all frames are unique,

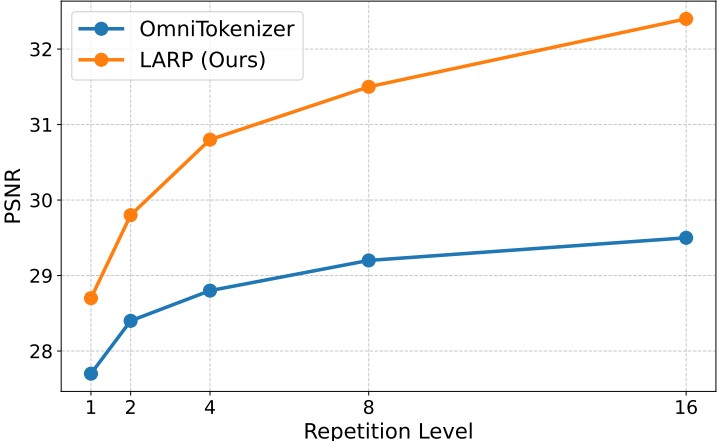

Figure 8: **Reconstruction PSNR across different repetition levels.**

while a value of 16 represents full repetition, where a single frame is repeated to create the entire video.

Figure 8 reports the reconstruction PSNR for various repetition levels. The results demonstrate the impact of increasing information redundancy on reconstruction quality for both OmniTokenizer and LARP. As the repetition level increases, both tokenizers show improved PSNR, reflecting their ability to leverage the added redundancy. However, LARP consistently outperforms OmniTokenizer across all repetition levels, with a significantly larger improvement in PSNR as the redundancy increases. These results validate LARP's effectiveness in capturing spatial-temporal redundancy, enabling it to represent videos more accurately within a compact latent space.

### B.3 ANALYSIS AND VISUALIZATION OF LARP'S LATENT SPACE

To gain deeper insights into the structure and properties of LARP's latent space, we randomly select 10 videos as examples (shown in Figure 9) and analyze the impact of individual LARP tokens on the reconstruction quality of these videos. Using the LARP-L-Long tokenizer, we first calculate the latent LARP tokens for all selected videos and measure their reconstruction PSNR values based on the unmodified LARP tokens. Next, for each video, we systematically modify a specific LARP token by assigning it a random value, then reconstruct the video and compute the PSNR again. The influence of the modified token on reconstruction quality is quantified by the difference between the PSNR values obtained from the unmodified and the single-token-modified reconstructions. This process is repeated for all 1024 LARP tokens across the latent space and for all 10 videos. The resulting PSNR differences are visualized in the scatter plots in Figure 10.

Interestingly, we observe that for each video, the majority of LARP tokens have minimal impact on reconstruction quality, while only a small subset of tokens carry the critical information necessary to represent the video accurately. However, the specific set of important tokens varies significantly across different videos, emphasizing the content-adaptive nature of LARP tokens, as different subsets of tokens are activated to represent different content.

Beyond the video-level analysis of LARP tokens conducted above, we also examine their impact at the spatial-temporal level. For LARP tokens of 5 different videos, we select a specific token index and set the values of tokens at this index in all videos to 0. We then compare the spatial-temporal reconstruction differences before and after this modification and visualize these differences as heatmaps, as shown in Figure 11. The brightness of the heatmaps have been enhanced for improved visibility.

It is evident that this token's influence spans the entire video, unlike patchwise tokens, which primarily affect specific spatial-temporal video patches. This finding confirms that LARP tokens serve as

holistic representations for videos. Another intriguing observation is that the LARP token's impact is not random; rather, it is concentrated on specific objects and structures, reflecting the semantic properties of the LARP's latent space.

## B.4 ANALYSIS OF LARP'S AUTOREGRESSIVE PRIOR

The AR prior learned by LARP plays a crucial role in its video generation performance, as shown in Figure 1 (b). In this section, we present additional analysis and visualizations to illustrate the properties of the learned AR prior.

**N-gram Histogram.** Given the discrete latent space of LARP , the AR prior is inherently abstract. To facilitate effective visualization, we analyze a simplified version of the AR prior, namely the n-gram prior, which represents the AR prior restricted to an n-token context window. Specifically, uniformly conditioned on all 101 classes, we sample one million token sequences from UCF101 videos using the LARP AR model and perform the same process with a baseline model that excludes the AR prior. We then count the occurrence frequencies of all n-grams across the video sequences. Finally, we plot histograms of the unique n-grams for both LARP and the no-prior baseline to compare their distributions.

Due to computational constraints, we perform this analysis for 2-grams and 3-grams. Figure 12 presents the 2-gram histograms for LARP and its no-prior baseline. Figure 13 provides a zoomed-in version of Figure 12, showing only 2-grams that occur fewer than 30k times. Figure 14 displays the 3-gram histograms. In all histograms, the x-axis represents the frequency, or the number of occurrences of a particular n-gram, while the y-axis represents the number of unique n-grams that occur with the given frequency across all sequences.

It is evident that compared to the histograms with the AR prior, those without the AR prior are significantly more left-skewed. This indicates that the distribution of the latent space without the AR prior is closer to a uniform distribution, where all n-grams have similar occurrence frequencies. In contrast, the histograms with the AR prior reveal that a small number of n-grams occur significantly more frequently than others, reflecting the autoregressive nature of the space they reside in.

**Class Dominance Score.** We define a class dominance score for 2-grams to compare the latent space with and without the AR prior. Specifically, using the sampled UCF101 video token sequences, we count the occurrence frequencies of all 2-grams for each UCF101 class. For every 2-grams $b_i$ that occurs more than a fixed threshold $d$ globally, we identify the UCF101 class $c_j$ it dominates, i.e., , the class to which the majority of occurrences of $b_i$ belong. The class dominance score quantifies the dominance of $b_i$ over its dominating class $c_j$.

Mathematically, the class dominance score $S(b_i)$ is defined as

$$S(b_i) = \max_{j \in \{1, \ldots, 101\}} \left( \frac{n(b_i, c_j)}{n(b_i)} \right),$$

(9)

where $n(b_i, c_j)$ denotes the number of occurrences of $b_i$ in video token sequences belonging to class $c_j$, and $n(b_i)$ represents the total number of occurrences of $b_i$ across all video token sequences.

After applying the threshold $d$, we obtain the set of class dominance scores:

$$\{S(b_i) \mid n(b_i) \geq d\},$$

(10)

where $n(b_i) \geq d$ ensures that only 2-grams with sufficient occurrences are included. This set is computed for both the latent space with and without the AR prior, then sorted and plotted in Figure 15. The x-axis denotes different 2-grams, and the y-axis represents the class dominance score.

It is worth noting that the area corresponding to the sequences without the AR prior is confined to a small region in the bottom-left corner. This indicates that most 2-grams in the no-prior sequences are uniformly distributed across all UCF101 classes. In contrast, the area for sequences with the AR prior is significantly larger, reflecting that many frequent 2-grams in these sequences are correlated with specific UCF101 classes. This significant disparity further highlights the impact of the AR prior.

## C    ADDITIONAL VISUALIZATION RESULTS

### C.1    VIDEO RECONSTRUCTION COMPARISON

Additional video reconstruction results are provided in Figure 16. Across a variety of scenes and regions, LARP consistently demonstrates superior reconstruction quality compared to OmniTokenizer Wang et al. (2024).

### C.2    CLASS-CONDITIONAL VIDEO GENERATION ON UCF-101 DATASET

We provide additional class-conditional video generation results in Figure 17. These results further demonstrate LARP's ability to generate high-quality videos with both strong per-frame fidelity and temporal consistency across various action classes in the UCF-101 dataset. The generated videos show diverse scene dynamics, capturing fine-grained details and natural motion, highlighting LARP's effectiveness in handling complex generative tasks within this challenging dataset. I

Generated video files (in MP4 format) are available in the supplementary materials.

### C.3    VIDEO FRAME PREDICTION ON K600 DATASET

We present additional video frame prediction results in Figure 18, further demonstrating LARP's capacity to accurately predict future frames in the K600 dataset. These results showcase LARP's ability to handle a wide range of dynamic scenes, capturing temporal dependencies with natural motion and smooth transitions between predicted frames. The predictions highlight LARP's effectiveness in scenarios involving complex motion and scene diversity, underscoring its strong generalization capabilities in video frame prediction tasks.

The predicted frames and the ground truth videos (in MP4 format) are available in the supplementary materials.

## D    LIMITATIONS

While LARP significantly enhances video generation quality, it still has certain limitations. Like other transformer-based video tokenizers Wang et al. (2024), LARP performs best with fixed-resolution videos due to the constraints of positional encoding. Additionally, artifacts may appear in LARP-generated videos when the scenes are particularly complex. Fortunately, scaling up the AR generative model is expected to improve video quality and reduce these artifacts, as suggested by the scaling laws of AR models Henighan et al. (2020); Sun et al. (2024).

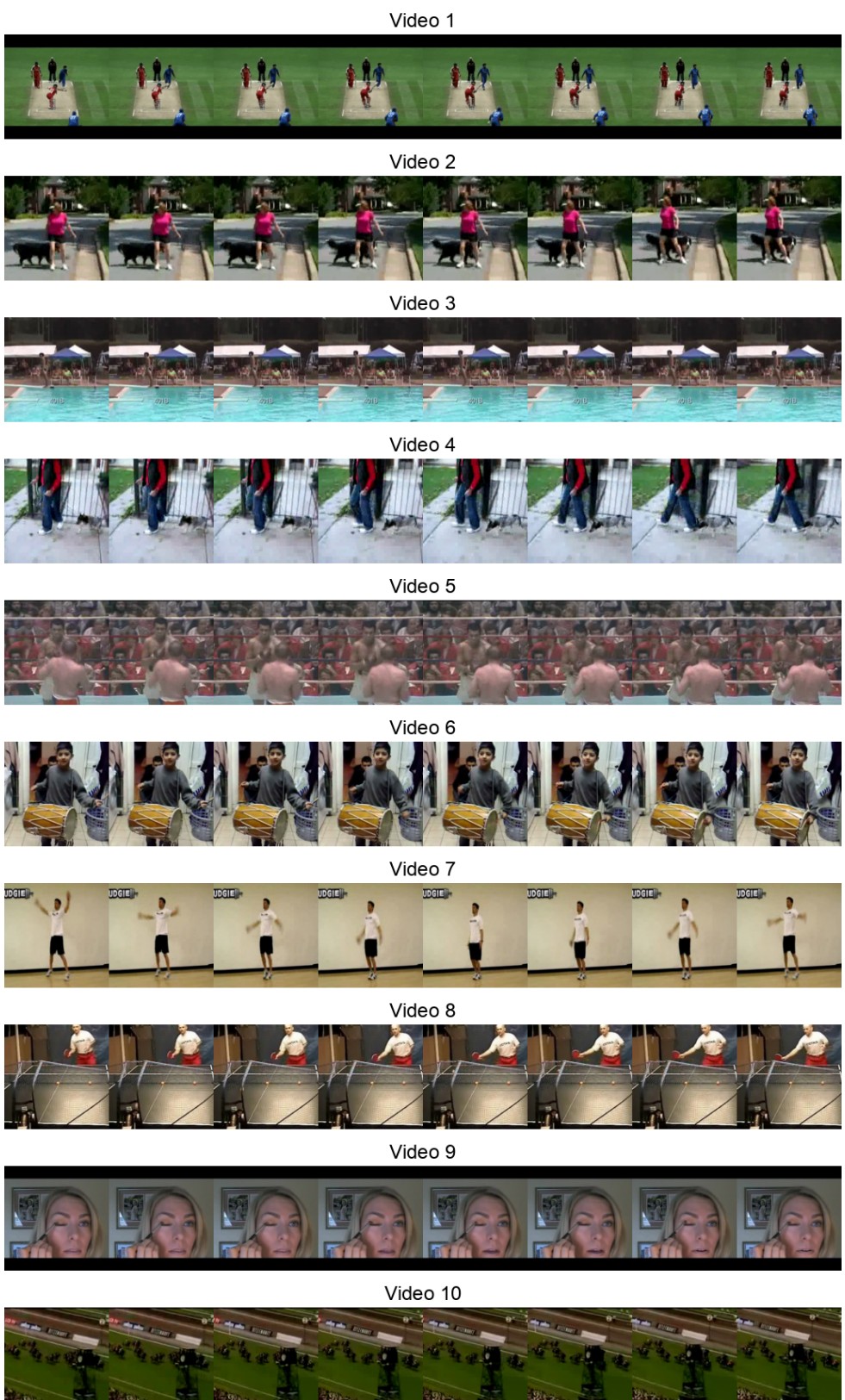

Figure 9: **Ground truth videos used for LARP token space analysis.**

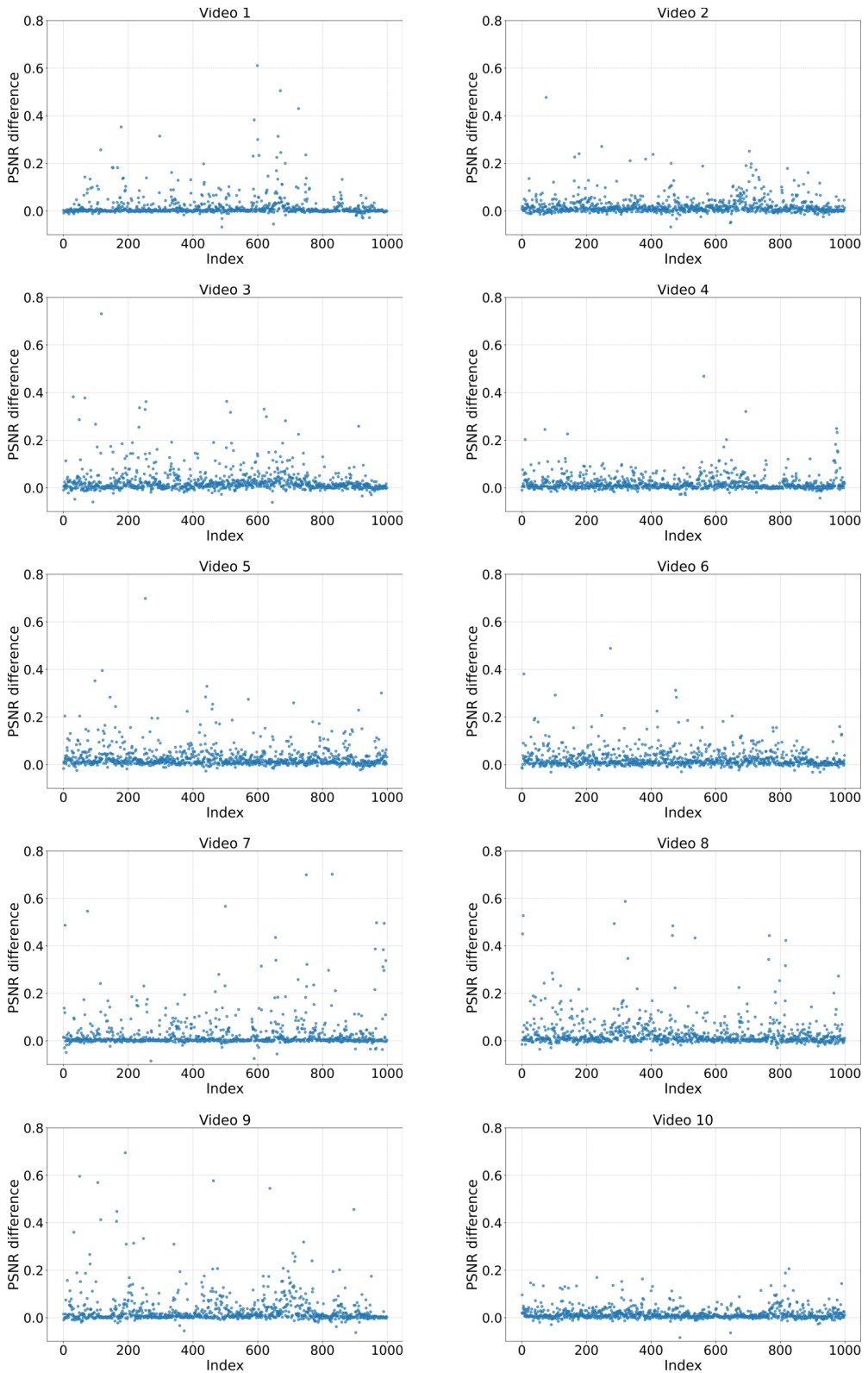

Figure 10: **PSNR differences induced by modifying individual LARP tokens.**

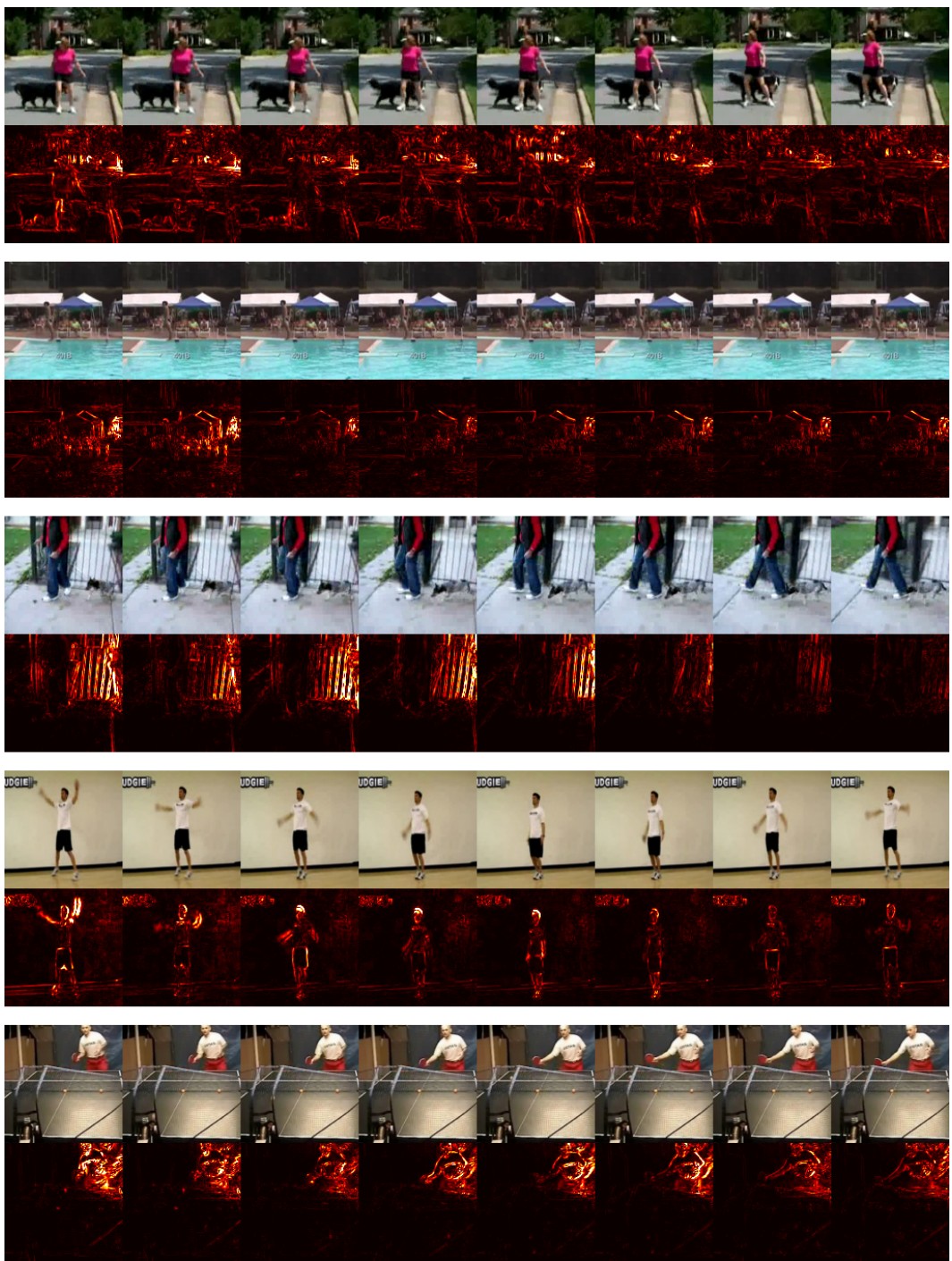

Figure 11: **Heatmap of reconstruction errors caused by modifying a single LARP token.** The tokens with the same specific index are modified across all 5 videos. In each group, the first row shows the ground truth video frames, while the second row displays the corresponding heatmaps.

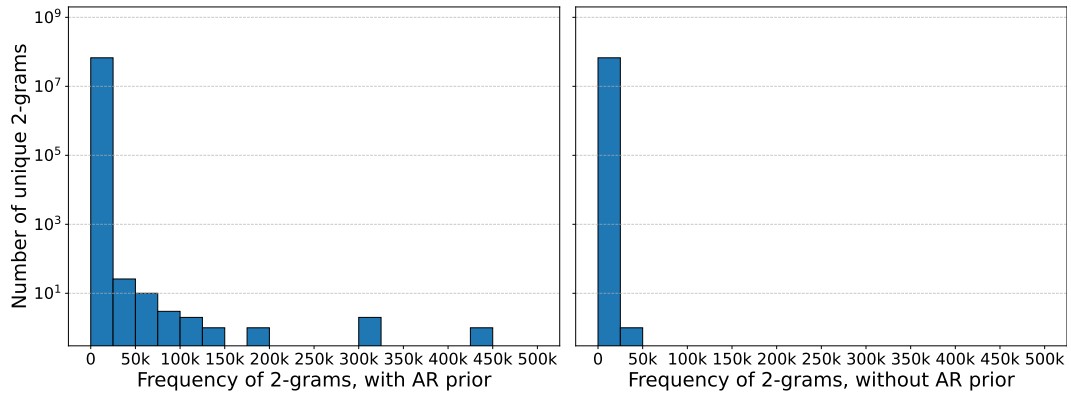

Figure 12: **Histogram of 2-grams.**

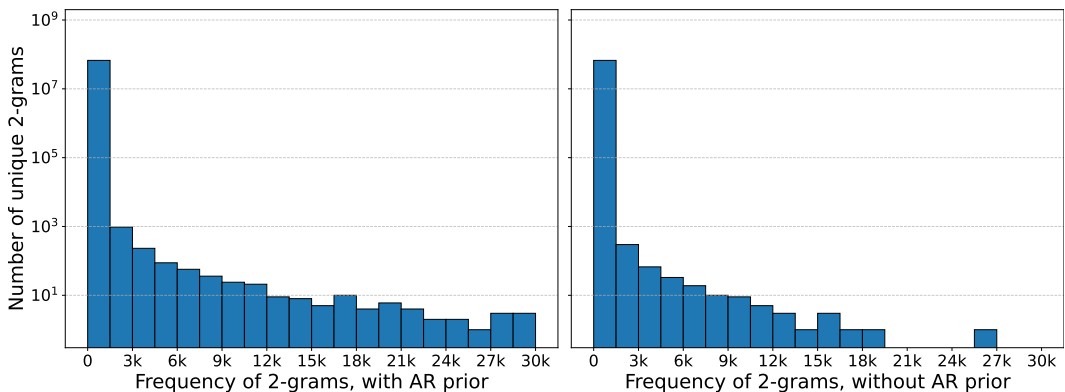

Figure 13: **Histogram of 2-grams (frequency $< $ 30k).**

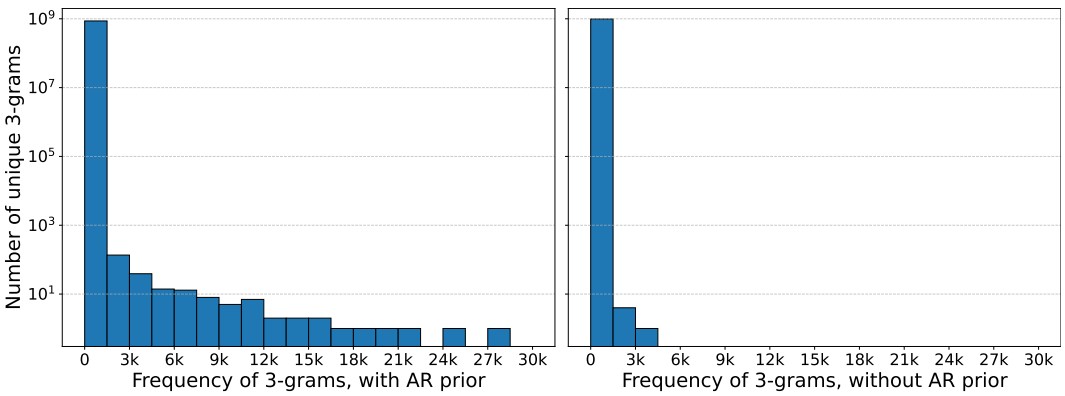

Figure 14: **Histogram of 3-grams.**

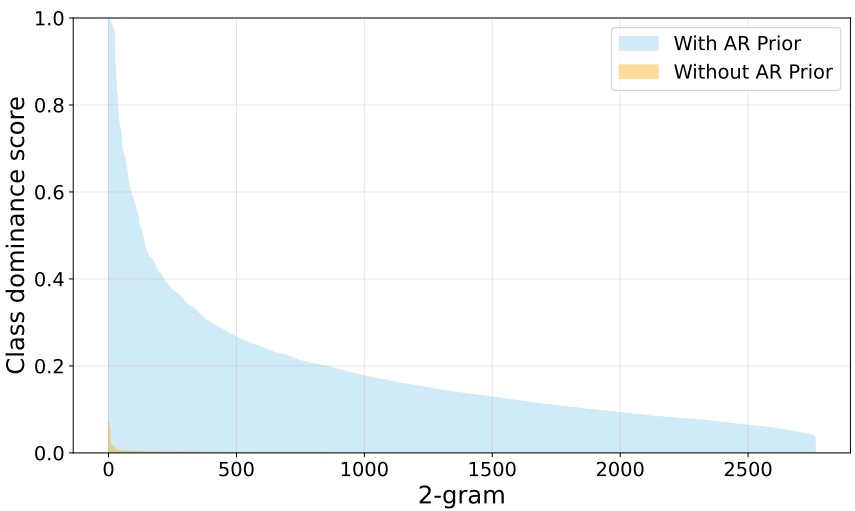

Figure 15: **Class dominance score.**

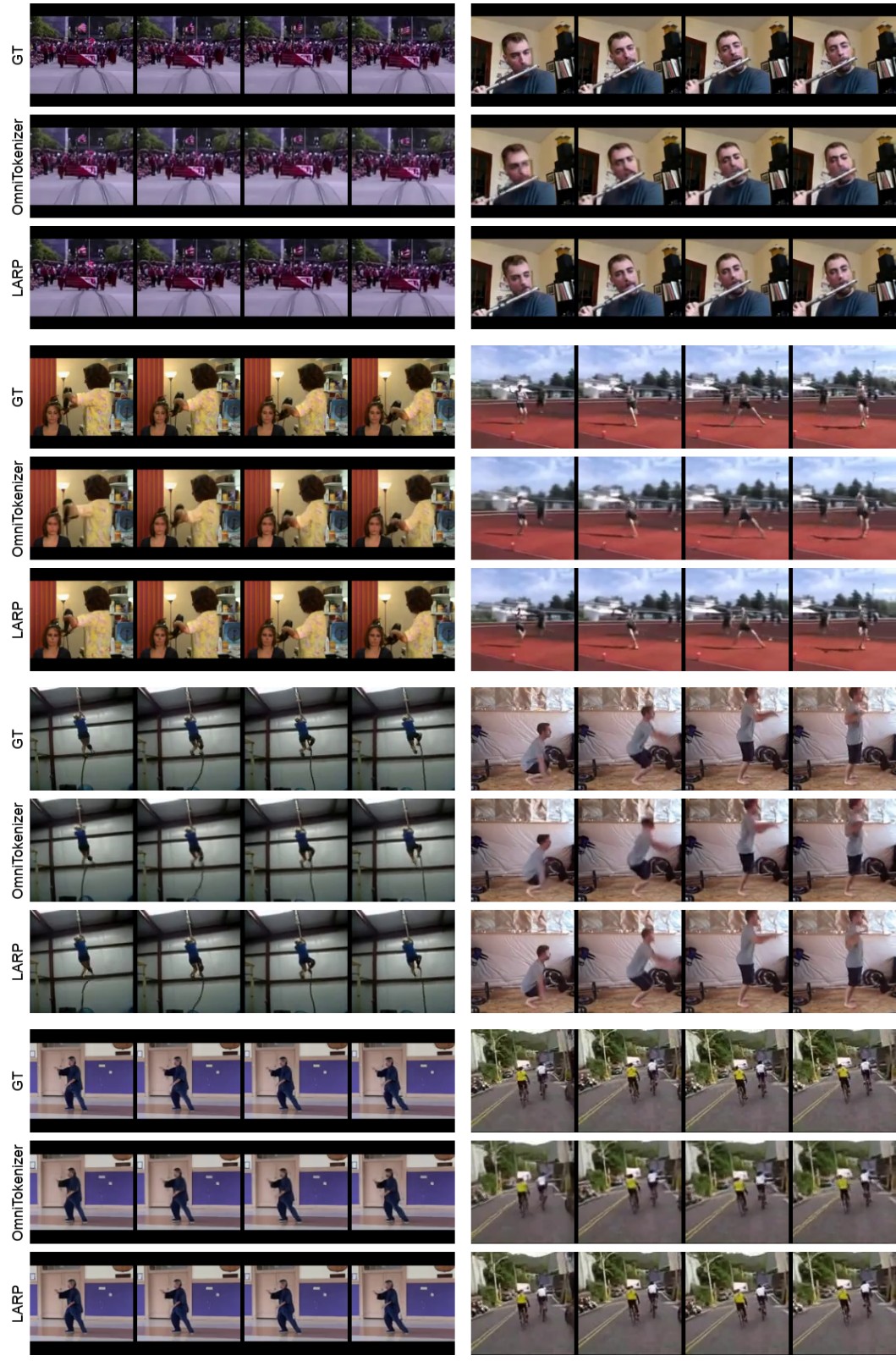

Figure 16: **Additional video reconstruction comparison** with OmniTokenizer (Wang et al., 2024).

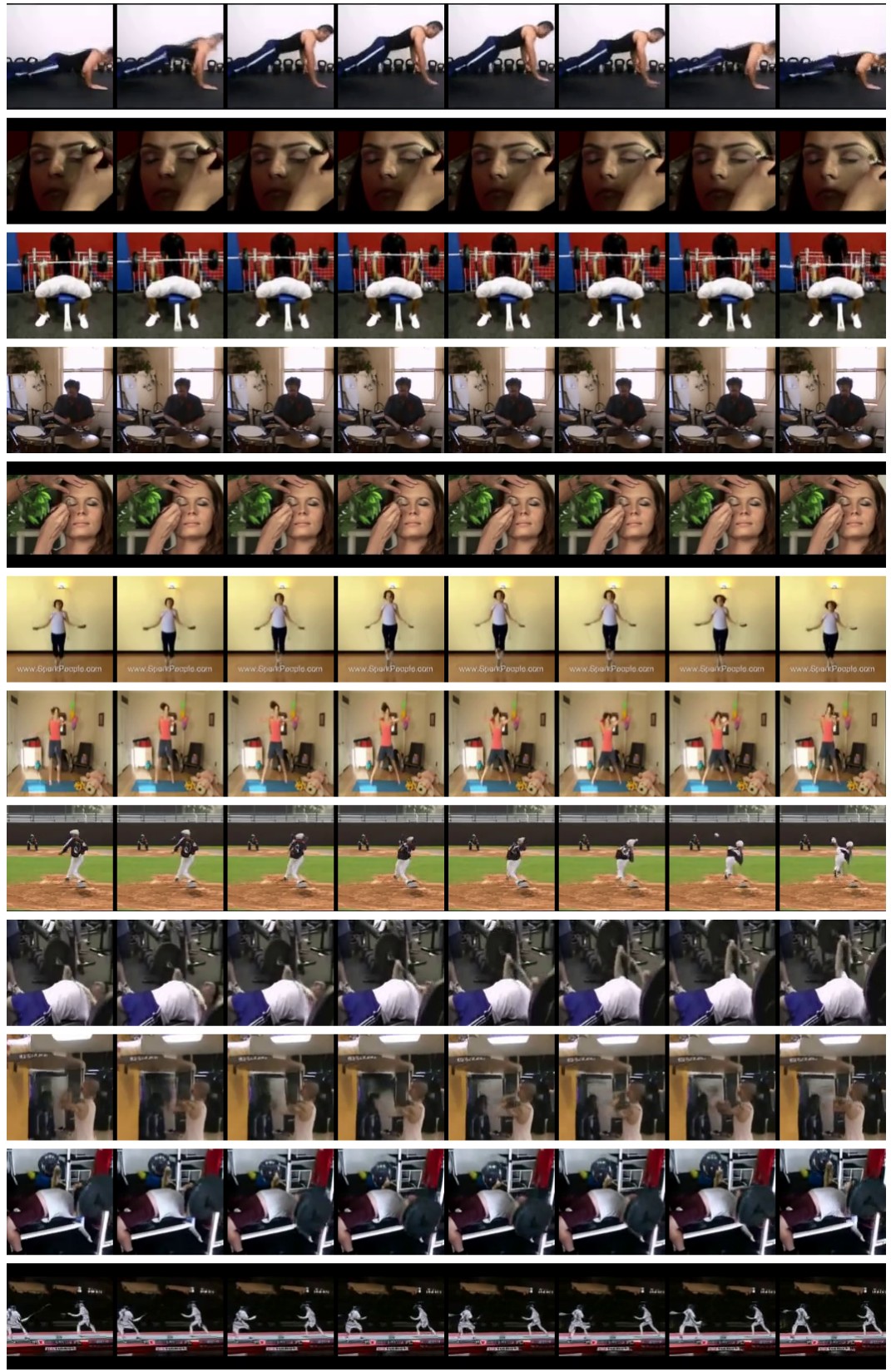

Figure 17: **Additional class-conditional generation results** on UCF-101 dataset.

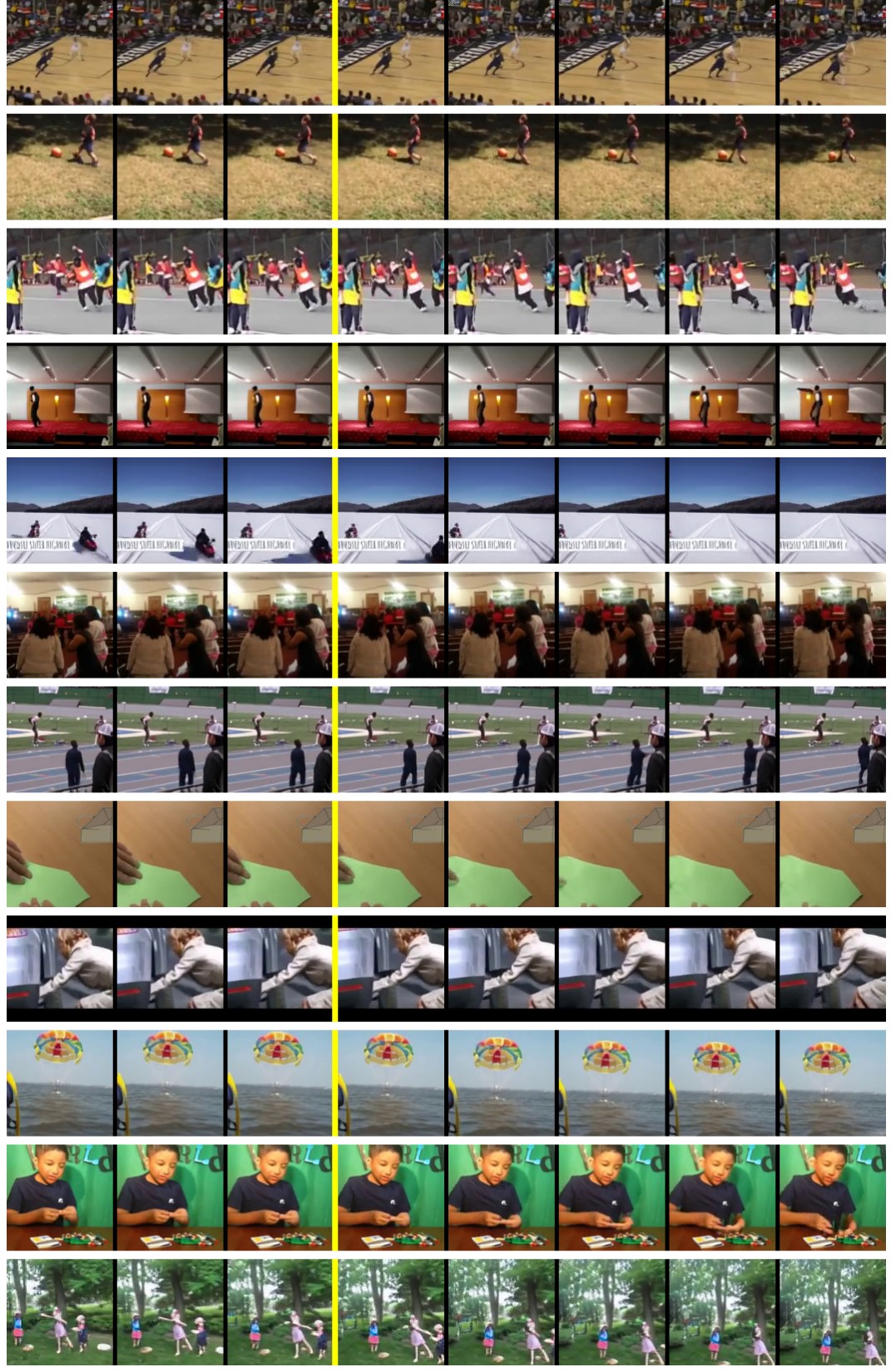

Figure 18: **Additional video frame prediction results** on K600 dataset.

