# OpenReview forum: "LARP: Tokenizing Videos with a Learned Autoregressive Generative Prior"
_ICLR.cc/2025/Conference — ICLR 2025 Oral_

### Official Review · Reviewer_cs2m · 2024-10-27

**Soundness:** 2
**Presentation:** 2
**Contribution:** 3
**Rating:** 8
**Confidence:** 3

**Summary:**

This paper proposes a video tokenizer for autoregressive (AR) generative models. The authors find that existing tokenizers use only patchwise tokenization without effective ordering, and face the design gap between reconstruction and AR generation. To address both limitations, they propose a new tokenizer with two key designs: (1) holistic tokenization that gathers information with learnable queries; (2) a lightweight AR transformer that optimizes the latent space for AR generation. Based on the proposed tokenizer, the authors trained a set of AR video generative models that achieve competitive performance on the Kinetics-600 and UCF-101 datasets.

**Strengths:**

* The paper is generally well written and easy to follow.
* The authors provide an insightful analysis of the two limitations of AR tokenization, and the proposed tokenizer with AR prior effectively addresses these limitations.
* Extensive quantitative and qualitative results are presented to support the effectiveness of the proposed method in video generation.

**Weaknesses:**

* The first contribution (holistic tokenization) is similar to TiTok [1] in that they both use learnable queries to tokenize the input data into a 1d sequence. This somewhat weakens the contribution of the paper. It would be better if the authors could detail their design differences in this part and justify the advantage of these differences through experiments.
* For the second contribution (AR prior model), there is no analysis of its accuracy or generalizability. Otherwise, it seems to me difficult for a small 21.7M AR model to predict the next tokens in a highly complex video latent space. The previous work VideoPoet [2] and Video-LaVIT [3] all require a 7B+ LLM to achieve this.
* Since the work focuses on generation performance, it would be better to introduce more metrics other than FVD, such as FID and IS. These metrics are also commonly used on UCF-101 and will significantly increase the soundness of the paper.
* There is a lack of intuitive analysis of the learned AR prior and latent space in the paper. Adding additional visualizations of the token ordering (e.g., by heatmap) would help readers understand how it actually works in the overall procedure. Otherwise, readers can only infer its functionality by looking at the performance improvement.

---

[1] Yu, et al. An Image is Worth 32 Tokens for Reconstruction and Generation. NeurIPS 2024.

[2] Kondratyuk, et al. VideoPoet: A Large Language Model for Zero-Shot Video Generation. ICML 2024

[3] Jin, et al. Video-LaVIT: Unified Video-Language Pre-training with Decoupled Visual-Motional Tokenization. ICML 2024

**Questions:**

* My main question after reading the paper is: The video data is already nicely ordered by timestamps, why not use this ordering directly for tokenization? Does the learned tokenizer reflect this natural order in its latent space? If not, I think the video tokenizer might be suboptimal.

---

> ### Author Response · Authors · 2024-11-18
> **Official Comment by Authors**
>
> We thank Reviewer cs2m for their insightful and constructive review of our paper. We appreciate the positive feedback on our analysis of AR tokenization limitations, the proposed holistic tokenizer, and the strong experimental results demonstrating its effectiveness. We address each of the raised concerns and suggestions in detail below.
>
> > W1: Comparison with [1].
>
> We would like to highlight that, as noted at the end of Section 2.1 in our original paper, [1] is a concurrent work to ours.
>
> Although [1] also uses holistic tokens to represent visual signals, LARP and [1] differ significantly in several key aspects, as outlined below:
>
> 1. **Order and Prior**: [1] does not define an order for their latent tokens and does not use prior models or regularization losses to enhance the performance of their downstream generative models. In contrast, the AR prior model is a core contribution of LARP, substantially enhancing video generation quality, as demonstrated in Figure 1(c) of our main paper.
> 1. **Generative Model**: [1] uses MaskGIT [5] as their video generator, a masked language modeling (MLM) generative model. In contrast, LARP is designed for AR generative models, known for their exceptional effectiveness and scalability.
> 1. **Focus**: [1] focuses on image generation, but LARP focuses on video generation.
> 1. **De-tokenization**: [1]'s ViT decoder operates similarly to MAE [2], where multiple ***identical mask tokens*** are input to their ViTs to reconstruct the images patches. In contrast, LARP is inspired by DETR [3] and the Q-former introduced in [4], where ***distinct query tokens*** are used to extract holistic tokens and reconstruct video patches.
> 1. **Training Strategy**: [1] employs a two-stage training process. In the first stage, they train their image tokenizer with discrete codes generated by an off-the-shelf MaskGIT-VQGAN model, akin to a model distillation approach. In the second stage, they fine-tune the decoder using the VQGAN objective. In contrast, the LARP tokenizer is trained end-to-end with a VQGAN-like objective, making it simpler and eliminating the need for other pre-trained models.
> 1. **Discriminator Design**: For the GAN loss, [1] uses a ConvNet-based discriminator, while LARP employs a transformer-based discriminator, enhancing reconstruction fidelity and training stability. We observed that training LARP with a ConvNet-based discriminator is not stable and often leads to loss divergence.
>
> In fact, we began this project well before the preprint of [1] was released.
>
>
> > W2: For the second contribution (AR prior model), there is no analysis of its accuracy or generalizability. Otherwise, it seems to me difficult for a small 21.7M AR model to predict the next tokens in a highly complex video latent space. The previous work VideoPoet [2] and Video-LaVIT [3] all require a 7B+ LLM to achieve this.
>
> We would like to clarify that the AR prior model is used exclusively during the training of the LARP tokenizer. Its role is to align the discrete latent space of LARP with AR generation tasks, leading to significantly improved generation performance, as demonstrated in Figure 1(b) of the original paper. For this purpose, a small AR model, like the one we used, is sufficient. Using a larger AR model would significantly slow down the training of the LARP tokenizer.
>
> It is important to note that the AR prior model is not involved in training the AR generator. As shown in Table 1 of the original paper, a 632M AR model is trained to achieve state-of-the-art FVD scores on the UCF101 class-conditional generation benchmark. This model size is appropriate given that UCF101 is a smaller-scale, academic-purpose video dataset, unlike the large-scale datasets used in [2] and [3].

---

> ### Author Response · Authors · 2024-11-18
> **Official Comment by Authors (Continued)**
>
> > W3: FID and IS on UCF101.
>
> The FID (Fréchet Inception Distance) measures the Fréchet Distance between the distributions of Inception v3 network features for real and generated images. It is important to emphasize that FID is a metric specifically designed for evaluating image generation, not video generation, and thus is not an appropriate metric for evaluating LARP.
> While some earlier video generation works report "FID" scores, these are essentially FVD (Fréchet Video Distance) scores calculated using an outdated video feature extractor, the C3D network [4]. In contrast, LARP and recent works, including all those compared in our main paper, utilize the modern version of FVD in Table 1 of the main paper, which employs the advanced I3D network [5] as the video feature extractor.
>
> The Inception Score (IS) is not a reliable metric for evaluating generative models, as discussed in [6]. Moreover, calculating IS for videos relies on the outdated C3D network [4]. For these reasons, more recent works [7, 8] have moved away from using this metric to evaluate video generators. Consequently, we did not include IS comparisons in our original paper. However, as requested, we provide a comparison of the IS of LARP with other works reporting IS in the table below:
>
> | Method             | Inception Score |
> | ------------------ | :-------------: |
> | VideoFusion [9]    |      80.03      |
> | HPDM [10]          |      87.68      |
> | CogVideo [11]      |      50.46      |
> | TATS [12]          |      79.28      |
> | MAGVIT [13]        |      89.27      |
> | LARP-L-Long (Ours) |    **90.04**    |
> | *GT*               |     *92.16*     |
>
> Here, *GT* denotes the IS calculated for ground truth UCF101 videos. Notably, LARP achieves the highest IS among all methods, with only a minimal gap compared to the ground truth videos. This comparison further highlights the exceptional video generation capabilities of LARP.
>
>
>
>
> > W4: Latent space analysis and visualization.
>
> We thank the reviewer for the suggestion. To gain deeper insights into the structure and properties of LARP's latent space, we randomly select 10 videos as examples (shown in Figure 9 of the revised paper) and analyze the impact of individual LARP tokens on the reconstruction quality of these videos.
>
> Using the LARP-L-Long tokenizer, we first calculate the latent LARP tokens for all selected videos and measure their reconstruction PSNR values based on the unmodified LARP tokens. Next, for each video, we systematically modify a specific LARP token by assigning it a random value, then reconstruct the video and compute the PSNR again. The influence of the modified token on reconstruction quality is quantified by the difference between the PSNR values obtained from the unmodified and the single-token-modified reconstructions. This process is repeated for all 1024 LARP tokens across the latent space and for all 10 videos. The resulting PSNR differences are visualized in the scatter plots in Figure 10 of the revised paper.
>
> Interestingly, we observe that for each video, the majority of LARP tokens have minimal impact on reconstruction quality, while only a small subset of tokens carry the critical information necessary to represent the video accurately. However, the specific set of important tokens varies significantly across different videos, emphasizing the content-adaptive nature of LARP tokens, as different subsets of tokens are activated to represent different content.
>
> Beyond the video-level analysis of LARP tokens conducted above, we also examine their impact at the spatial-temporal level.
>
> For LARP tokens of 5 different videos, we select a specific token index and set the values of tokens at this index in all videos to 0. We then compare the spatial-temporal reconstruction differences before and after this modification and visualize these differences as heatmaps, as shown in Figure 11 of the revised paper. The brightness of the heatmaps has been enhanced for improved visibility.
>
> It is evident that this token's influence spans the entire video, unlike patchwise tokens, which primarily affect specific spatial-temporal video patches. This finding confirms that LARP tokens serve as holistic representations for videos.
> Another intriguing observation is that the LARP token's impact is not random; rather, it is concentrated on specific objects and structures, reflecting the semantic properties of the LARP latent space.
>
> This discussion has been incorporated into Section B.3 of the revised paper.

---

> ### Author Response · Authors · 2024-11-18
> **Official Comment by Authors (Continued)**
>
> > Q1: The video data is already nicely ordered by timestamps, why not use this ordering directly for tokenization? Does the learned tokenizer reflect this natural order in its latent space?
>
> Using timestamp order to tokenize videos is possible; however, this approach implies per-frame tokenization, which cannot effectively capture the temporal redundancy in videos, compromising the compactness of the latent space. Moreover, even if timestamp order is used, individual video frames cannot typically be represented with a single token, necessitating further decisions on how to tokenize the content within each frame.
>
> That said, the concept of timestamp order tokenization is compatible with LARP and could play a significant role in future work when scaling up to longer videos. In such cases, LARP could transition from per-video tokenization to per-clip tokenization, where the latent tokens of a long video would consist of multiple groups of LARP tokens for its clips, concatenated along the timestamp dimension. We are excited to explore this promising direction in future work.
>
>
>
> [1] Yu, et al. An Image is Worth 32 Tokens for Reconstruction and Generation. NeurIPS 2024.
>
> [2] Kondratyuk, et al. VideoPoet: A Large Language Model for Zero-Shot Video Generation. ICML 2024
>
> [3] Jin, et al. Video-LaVIT: Unified Video-Language Pre-training with Decoupled Visual-Motional Tokenization. ICML 2024
>
> [4] Tran, Du, et al. Learning spatiotemporal features with 3d convolutional networks. Proceedings of the IEEE international conference on computer vision. 2015.
>
> [5] Carreira, Joao, and Andrew Zisserman. Quo vadis, action recognition? a new model and the kinetics dataset. proceedings of the IEEE Conference on Computer Vision and Pattern Recognition. 2017.
>
> [6] Barratt, Shane, and Rishi Sharma. A note on the inception score. arXiv preprint arXiv:1801.01973 (2018).
>
> [7] Yu, Lijun, et al. Language Model Beats Diffusion-Tokenizer is key to visual generation. ICLR 2024.
>
> [8] Wang, Junke, et al. OmniTokenizer: A Joint Image-Video Tokenizer for Visual Generation. NeurIPS 2024.
>
> [9] Luo, Zhengxiong, et al. Videofusion: Decomposed diffusion models for high-quality video generation. arXiv preprint arXiv:2303.08320 (2023).
>
> [10] Skorokhodov, Ivan, et al. Hierarchical Patch Diffusion Models for High-Resolution Video Generation. Proceedings of the IEEE/CVF Conference on Computer Vision and Pattern Recognition. 2024.
>
> [11] Hong, Wenyi, et al. Cogvideo: Large-scale pretraining for text-to-video generation via transformers. arXiv preprint arXiv:2205.15868 (2022).
>
> [12] Ge, Songwei, et al. Long video generation with time-agnostic vqgan and time-sensitive transformer. European Conference on Computer Vision. Cham: Springer Nature Switzerland, 2022.
>
> [13] Yu, Lijun, et al. Magvit: Masked generative video transformer. Proceedings of the IEEE/CVF Conference on Computer Vision and Pattern Recognition. 2023.
>
>
>
> **We would like to thank Reviewer cs2m once again for their thoughtful review, which has greatly contributed to improving our work. If there are any additional concerns, we would be happy to address them. We respectfully request Reviewer cs2m to consider a higher rating in light of our detailed responses and revisions.**

---

> > ### Comment · Reviewer_cs2m · 2024-11-19
> >
> > Thanks for the additional experiments, which addressed most of my concerns. My remaining concern is that the visualizations for W4 (Figures 9-11) are not very informative; a model without the AR prior can have a similar heatmap. Given that the AR prior is a very abstract concept, could the authors think of specific ways to visualize/verify it?

---

> > > ### Author Response · Authors · 2024-11-22
> > >
> > > We thank Reviewer cs2m for their valuable suggestions. As noted by Reviewer cs2m, the AR prior is a highly abstract concept. To effectively analyze and visualize it, we propose two methods below:
> > >
> > > ## N-gram Histogram
> > >
> > > To facilitate effective visualization, we analyze a simplified version of the AR prior, namely the n-gram prior, which represents the AR prior restricted to an n-token context window.
> > >
> > > Specifically, uniformly conditioned on all 101 classes, we sample one million token sequences from UCF101 videos using the LARP AR model and perform the same process with a baseline model that excludes the AR prior. We then count the occurrence frequencies of all n-grams across the video sequences. Finally, we plot histograms of the unique n-grams for both LARP and the no-prior baseline to compare their distributions.
> > >
> > > Due to computational constraints, we perform this analysis for 2-grams and 3-grams. Figure 12 of the revised paper presents the 2-gram histograms for LARP and its no-prior baseline. Figure 13 of the revised paper provides a zoomed-in version of Figure 12, showing only 2-grams that occur fewer than 30k times. Figure 14 of the revised paper displays the 3-gram histograms.
> > > In all histograms, the x-axis represents the frequency, or the number of occurrences of a particular n-gram, while the y-axis represents the number of unique n-grams that occur with the given frequency across all sequences.
> > >
> > > It is evident that compared to the histograms with the AR prior, those without the AR prior are significantly more left-skewed. This indicates that the distribution of the latent space without the AR prior is closer to a uniform distribution, where all n-grams have similar occurrence frequencies. In contrast, the histograms with the AR prior reveal that a small number of n-grams occur significantly more frequently than others, reflecting the autoregressive nature of the space they reside in.
> > >
> > > ## Class Dominance Score
> > >
> > > We define a **class dominance score** for 2-grams to compare the latent space with and without the AR prior. Specifically, using the sampled UCF101 video token sequences, we count the occurrence frequencies of all 2-grams for each UCF101 class. For every 2-gram $b_i$ that occurs more than a fixed threshold $d$ globally, we identify the UCF101 class $c_j$ it dominates, i.e., the class to which the majority of occurrences of $b_i$ belong. The class dominance score quantifies the dominance of $b_i$ over its dominating class $c_j$.
> > >
> > > Mathematically, the class dominance score $S(b_i)$ is defined as:
> > >
> > > $$
> > > S(b_i) = \max_{j \in \{1, \dots, 101\}} \left( \frac{n(b_i, c_j)}{n(b_i)} \right),
> > > $$
> > >
> > > where:
> > > - $n(b_i, c_j)$: the number of occurrences of $b_i$ in video token sequences belonging to class $c_j$,
> > > - $n(b_i)$: the total number of occurrences of $b_i$ across all video token sequences.
> > >
> > > After applying the threshold $d$, we obtain the set of class dominance scores:
> > >
> > > $$
> > > \{ S(b_i) \mid n(b_i) \geq d \},
> > > $$
> > >
> > > where $n(b_i) \geq d$ ensures that only 2-grams with sufficient occurrences are included. This set is computed for both the latent space with and without the AR prior, then sorted and plotted in Figure 15 of the revised paper. The x-axis denotes different 2-grams, and the y-axis represents the class dominance score.
> > >
> > > It is worth noting that the area corresponding to the sequences without the AR prior is confined to a small region in the bottom-left corner. This indicates that most 2-grams in the no-prior sequences are uniformly distributed across all UCF101 classes. In contrast, the area for sequences with the AR prior is significantly larger, reflecting that many frequent 2-grams in these sequences are correlated with specific UCF101 classes. This significant disparity further highlights the impact of the AR prior.
> > >
> > > This discussion has been added to Section B.4 of the revised paper.

---

> > > > ### Comment · Reviewer_cs2m · 2024-11-24
> > > >
> > > > Thank you for the additional effort to make AR prior more intuitive. I have no further concerns and raise my score to 8.

---

> > > > > ### Author Response · Authors · 2024-11-25
> > > > >
> > > > > We deeply appreciate your valuable suggestions and thoughtful engagement with our responses! Your positive feedback and increased rating are a significant affirmation of our work.

---

### Official Review · Reviewer_3WZp · 2024-11-04

**Soundness:** 3
**Presentation:** 3
**Contribution:** 3
**Rating:** 6
**Confidence:** 4

**Summary:**

This paper presented a novel video tokenization technique for autoregressive video generation. It leveraged several query embeddings to gather information from input videos, which enabled the tokenizer to capture more global representations and support arbitrary number of tokens. It further integrated a lightweight autoregressive transformer model as a prior during training of the tokenizer, which can result in a better latent space for downstream autoregressive video generation. This paper also presented experimental results on the effectiveness of the proposed tokenizer when applied to video generation tasks.

**Strengths:**

- This paper presented an inspiring message that properly considering the downstream video generation tasks when training video tokenizers can result in much better generation performance.
- Extensive experiments and ablations demonstrated the effectiveness of incorporating the autoregressive prior to tokenizer training.

**Weaknesses:**

- There are several typos in the paper:
    - line 270: “back to the continues” -> “back to the continuous”
    - line 459: “hilighting” → “highlighting”
- More ablation studies can be conducted to analyze the effectiveness of holistic video tokenization and autoregressive generation further, as detailed in the Questions section.

**Questions:**

- As claimed and validated in the paper, an autoregressive prior is critical to improve generation performance. In this paper, it is implemented by leveraging an additional autoregressive transformer model. As presented in line 251, this paper used a transformer encoder architecture as a video tokenizer. If this transformer encoder is replaced by an encoder-decoder structure, will it serve as an autoregressive prior?
- Could the authors further explain why an autoregressive prior during tokenizer training can lead to better generation performance? An argument is that, lower autoregressive prior loss does not necessarily lead to better generation performance, since in an extreme case, if all the tokens are the same, then the autoregressive prior loss can be extremely low, but the codebook will collapse and the downstream generation task will definitely fail.
- This paper introduced the holistic video tokenization technique implemented by query-based transformers. Without this holistic video tokenization technique, the autoregressive prior can also be applied to the “patch tokens” as in Figure 2 (a). Despite the flexibility of supporting an arbitrary number of discrete tokens, the claim of “global and semantic representations” seems not validated in the paper. Are there experimental results supporting this claim?
- Table 1 shows the reconstruction comparison among different tokenizers. With 1280 tokens, the rFVD of MAGVIT-v2 is 8.6. With 1024 tokens, the rFVD of LARP-L-Long is 20. Since the tokenizer presented in the paper can support an arbitrary number of discrete tokens, what will the rFID be if 1280 tokens are used?

---

> ### Author Response · Authors · 2024-11-18
> **Official Comment by Authors**
>
> We thank Reviewer 3WZp for their constructive feedback and acknowledgment of our methodological and experimental contributions. We appreciate the thoughtful suggestions, which will help us improve our work further. We respond to each of the concerns raised below.
>
> > W1: Typos.
>
> We have corrected the typos in the revised paper.
>
> > W2: Effectiveness of holistic video tokenization and autoregressive generation, detailed in the Questions section.
>
> We address each question below.
>
> > Q1: If this transformer encoder is replaced by an encoder-decoder structure, will it serve as an autoregressive prior?
>
> We emphasize that the learned autoregressive prior is agnostic to model architecture details. It operates in the discrete latent space to align it with the downstream autoregressive generation task, without imposing assumptions on how data is encoded into this space.
> Thus, replacing the transformer encoder with an encoder-decoder structure would not affect the applicability of the learned autoregressive prior.
>
> In fact, we explored the encoder-decoder architecture during early experiments and observed the similar performance.
> However, the encoder-decoder structure introduces additional cross-attention layers, increasing model complexity and reducing parameter efficiency. For these reasons, we opted for the simpler transformer encoder architecture.
>
> > Q2: Could the authors further explain why an autoregressive prior during tokenizer training can lead to better generation performance? An argument is that, lower autoregressive prior loss does not necessarily lead to better generation performance, since in an extreme case, if all the tokens are the same, then the autoregressive prior loss can be extremely low, but the codebook will collapse and the downstream generation task will definitely fail.
>
>
> The extreme case mentioned in Q2 is a valid concern. To prevent such collapse, we combine the AR prior loss with the reconstruction loss and use a small weight $\alpha$ for the AR prior loss, as detailed in Eq. 8 and line 406 of the original paper.
> This ensures the reconstruction loss dominates the training objective, preserving the integrity and effectiveness of the codebook. This is why LARP achieves strong performance in both reconstruction and generation tasks, as demonstrated in Table 1 of the original paper.
>
> Learning an autoregressive prior during tokenizer training improves generation performance by aligning the tokenizer's discrete latent space with the autoregressive generation task. This alignment is achieved through the autoregressive prior loss, which encourages a more structured latent space, simplifying downstream autoregressive modeling.

---

> ### Author Response · Authors · 2024-11-18
> **Official Comment by Authors (Continued)**
>
> > Q3: Despite the flexibility of supporting an arbitrary number of discrete tokens, the claim of “global and semantic representations” seems not validated in the paper. Are there experimental results supporting this claim?
>
> The holistic tokenization scheme of LARP enables global and semantic representations, effectively capturing the spatiotemporal redundancy in videos and representing them accurately within a compact latent space.
> The semantic properties of LARP tokens are demonstrated by their exceptional generation quality.
> Moreover, we provide an extensive analysis and visualization of LARP's latent space in Section B.3 of the revised paper.
> Our visualizations reveal that LARP tokens influence the entire video, unlike patchwise tokens that primarily affect specific spatial-temporal patches. This finding underscores that LARP tokens act as holistic representations for videos. Interestingly, the impact of LARP tokens is not random; instead, it is focused on specific objects and structures, highlighting the semantic nature of the LARP latent space.
> We kindly refer you to Section B.3 for more details regarding the representations learned by LARP.
>
> To further validate LARP's capability to represent videos globally and holistically, we manually introduce information redundancy into videos and measure the improvement in reconstruction quality.
>
> Specifically, we fix the total number of video frames to 16, consistent with our main experiments, and repeat shorter videos multiple times to form 16-frame videos. This increases information redundancy as the number of unique frames decreases with higher repetition, implying improved reconstruction quality for tokenizers capable of exploiting global and holistic information.
>
> The table below reports the reconstruction PSNR for various repetition levels. We also compare LARP against a patchwise video tokenizer, OmniTokenizer.
> Notably, although OmniTokenizer operates in a patchwise manner, its transformer architecture allows it to leverage some global information.
> For repetition levels, a value of 1 indicates no repetition, where all frames are unique, while a value of 16 represents full repetition, where a single frame is repeated to create the entire video.
>
>
>
> | Repetition Level | 1    | 2           | 4           | 8           | 16          |
> | ---------------- | ---- | ----------- | ----------- | ----------- | ----------- |
> | OmniTokenizer    | 27.7 | 28.4 (+0.7) | 28.8 (+1.1) | 29.2 (+1.5) | 29.5 (+1.8) |
> | LARP (Ours)      | 28.7 | 29.8 (+1.1) | 30.8 (+2.1) | 31.5 (+2.8) | 32.4 (+3.7) |
>
> The results in the table demonstrate the impact of increasing information redundancy on reconstruction quality for both OmniTokenizer and LARP. As the repetition level increases, both tokenizers show improved PSNR, reflecting their ability to leverage the added redundancy. However, LARP consistently outperforms OmniTokenizer across all repetition levels, with a significantly larger improvement in PSNR as the redundancy increases.
>
> These results validate LARP's effectiveness in capturing spatial-temporal redundancy, enabling it to represent videos more accurately within a compact latent space.
>
> This discussion has been incorporated into Section B.2 of the revised paper, with the results from the table above visualized in Figure 8.

---

> ### Author Response · Authors · 2024-11-18
> **Official Comment by Authors (Continued)**
>
> > Q4: What will the rFVD be if 1280 tokens are used?
>
>
> We evaluate the rFVD of LARP-L-Long with 1280 tokens and compare it with the rFVD scores of other methods and configurations, as shown in the table below:
>
>
> | Method                 | #Tokens | rFVD  |
> |------------------------|:-------:|:-----:|
> | TATS [1]               | 1024    | 162   |
> | MAGVIT [2]             | 1024    | 25    |
> | MAGVIT-v2 [3]          | 1280    | 8.6   |
> | OmniTokenizer [4]      | 1280    | 42    |
> | LARP-L (Ours)          | 1024    | 24    |
> | LARP-L-Long (Ours)     | 1024    | 20    |
> | LARP-L-Long-1280 (Ours)| 1280    | 16    |
>
>
> It is evident that increasing the number of tokens used by LARP from 1024 to 1280 leads to a significant improvement in rFVD. The use of 1280 tokens enhances the information capacity of LARP's latent space, resulting in better reconstruction quality.
>
>
> [1] Ge, Songwei, et al. Long video generation with time-agnostic vqgan and time-sensitive transformer. European Conference on Computer Vision. Cham: Springer Nature Switzerland, 2022.
>
> [2] Yu, Lijun, et al. Magvit: Masked generative video transformer. Proceedings of the IEEE/CVF Conference on Computer Vision and Pattern Recognition. 2023.
>
> [3] Yu, Lijun, et al. Language Model Beats Diffusion-Tokenizer is key to visual generation. ICLR 2024.
>
> [4] Wang, Junke, et al. OmniTokenizer: A Joint Image-Video Tokenizer for Visual Generation. NeurIPS 2024.
>
>
>
>
>
>
> **We sincerely thank reviewer 3WZp once again for their valuable feedback, which has played a crucial role in improving our work. If there are any additional concerns, we would be happy to address them. We are committed to ensuring clarity and providing comprehensive responses to all concerns. We kindly request reviewer 3WZp to consider a higher rating in light of the revisions and responses provided.**

---

> ### Comment · Reviewer_3WZp · 2024-11-26
>
> Thanks for the authors’ response. I still have some questions.
>   - By increasing the number of tokens from 1024 to 1280, we can indeed observe better performance in rFVD. However, there is still a gap between MAGVIT-v2 (16 vs 8.6). Where do you think the gap comes from?
>   - I’m still wondering if the holistic video tokenization technique is necessary for incorporating with the autoregressive prior. Can the autoregressive prior be added to patchwise tokens and lead to improved generation performance?
>   - For the comparison in Section B.3, I acknowledge the authors’ effort in demonstrating the advantage of holistic video tokenization over patchwise video tokenizers for better leveraging redundancy. However, since MAGVIT-v2 is a much stronger baseline than OmniTokenizer, I’m still curious how MAGVIT-v2 will perform in this setting.

---

> > ### Author Response · Authors · 2024-11-27
> >
> > Thank you for your additional questions. Please find our responses to each of them below.
> >
> > > AQ1: By increasing the number of tokens from 1024 to 1280, we can indeed observe better performance in rFVD. However, there is still a gap between MAGVIT-v2 (16 vs 8.6). Where do you think the gap comes from?
> >
> > We emphasize that LARP is specifically designed and optimized for video generation. As shown in Table 2 of the original paper, design choices such as `No AR prior model`, `No scheduled sampling`,   `Deterministic quantization` are all tailored to improve gFVD at the expense of rFVD.
> > During the development of LARP, we intentionally prioritized generation quality over reconstruction quality, as achieving superior generation performance is the ultimate goal and motivation of our work.
> >
> > In contrast, the MAGVIT-v2 tokenizer [1] is primarily designed and optimized for video tokenization. As evidenced by their ablation studies (Table 5(c) in [1]), all design choices are validated to enhance tokenization quality rather than generation quality, as indicated by their focus on reconstruction metrics such as rFVD and LPIPS.
> >
> > As a result, MAGVIT-v2 achieves better rFVD compared to LARP. However, its gFVD is inferior to LARP's, particularly when applied to AR generative models. In summary, the rFVD gap between MAGVIT-v2 and LARP arises from the differing focuses of their respective design choices.
> >
> > > AQ2: I'm still wondering if the holistic video tokenization technique is necessary for incorporating with the autoregressive prior. Can the autoregressive prior be added to patchwise tokens and lead to improved generation performance?
> >
> > LARP leverages holistic video tokenization for its semantic richness and flexibility, supporting an arbitrary number of video tokens. This adaptability allows downstream applications to balance representation size and reconstruction quality, highlighting its potential for tokenizing real-world videos when integrated with multimodal large language models (mLLMs).
> >
> > However, a standalone holistic tokenizer lacks a pre-defined token order, which is essential for autoregressive (AR) generation tasks. To address this, our proposed AR prior aligns the latent space with the task of AR generation and defines a learned order within the token space. This approach significantly improves video generation quality, as demonstrated in Figure 1(b) of our paper, underscoring the essential role of the learned AR prior in enhancing holistic tokenization.
> >
> > While the learned AR prior is specifically designed to optimize our holistic video tokenizer, its applications are not restricted to this paradigm. Fundamentally, the learned AR prior acts as a regularization mechanism for the discrete latent space of an autoencoder, without requiring the latent space to adhere to a specific encoding scheme, such as holistic or patchwise tokenization.
> >
> > Therefore, we believe the learned AR prior can also improve the performance of patchwise video tokenizers, such as OmniTokenizer [2]. However, incorporating it into such frameworks may require further refinements to module designs and tuning of hyperparameters to accommodate differences in model architecture and latent space structure, ensuring proper compatibility.
> >
> >
> >
> > > AQ3: For the comparison in Section B.3, I acknowledge the authors' effort in demonstrating the advantage of holistic video tokenization over patchwise video tokenizers for better leveraging redundancy. However, since MAGVIT-v2 is a much stronger baseline than OmniTokenizer, I'm still curious how MAGVIT-v2 will perform in this setting.
> >
> > Thank you for acknowledging our effort. We highlight that while MAGVIT-v2 [1] is indeed a stronger baseline than OmniTokenizer [2], it is a closed-source, proprietary model. Even unofficial third-party implementations, such as [3], are limited to image tokenization and generation tasks, making them unsuitable for direct comparison with LARP's focus on video generation.
> >
> > As a result, we can only compare LARP with MAGVIT-v2 using the metrics officially reported in their paper. Due to the lack of access to their pre-trained models, we are unable to conduct additional comparisons involving MAGVIT-v2.
> >
> >
> >
> > [1] Yu, Lijun, et al. Language Model Beats Diffusion-Tokenizer is key to visual generation. ICLR 2024.
> >
> > [2] Wang, Junke, et al. OmniTokenizer: A Joint Image-Video Tokenizer for Visual Generation. NeurIPS 2024.
> >
> > [3] Luo, Zhuoyan, et al. Open-magvit2: An open-source project toward democratizing auto-regressive visual generation. arXiv preprint arXiv:2409.04410 (2024).

---

> > > ### Author Response · Authors · 2024-12-02
> > >
> > > We sincerely value your thoughtful feedback and deeply appreciate your additional questions. We hope our responses address your concerns effectively. If you feel that increasing your score is warranted, we would be truly grateful. Additionally, please don’t hesitate to share any further questions or comments—your perspective is extremely important to us.
> > >
> > > Thank you once again for your time and careful consideration.

---

### Official Review · Reviewer_qHTB · 2024-11-04

**Soundness:** 3
**Presentation:** 3
**Contribution:** 3
**Rating:** 8
**Confidence:** 5

**Summary:**

The submission introduces LARP, a novel video tokenizer designed to optimize video tokenization for autoregressive (AR) generative models. The approach deviates from traditional patch-wise tokenizers by adopting a holistic strategy, which captures global and semantic representations of video content. This unique structure allows for adaptive tokenization, with a focus on enhancing AR compatibility through an integrated lightweight AR prior model. The experimental results show state-of-the-art performance in video generation benchmarks, particularly in UCF-101 and K600 datasets, establishing the proposed method as a competitive alternative for high-fidelity video synthesis.

**Strengths:**

* The proposed method addresses an interesting and important topic with great potential for multimodal LLMs - how to tokenize videos into a sequence of tokens more suitable for LLM learning.
* The introduced AR prior model is a simple yet effective method to produce tokens more friendly for autoregressive generation.
* Experiment results demonstrate the effectiveness of the proposed method in reducing the gap between generation quality and reconstruction quality, highlighting better learnability of the produced tokens.
* State-of-the-art performance is presented on the UCF benchmark, with K600 results outperforming AR baselines and ablation studies verifying key design choices.

**Weaknesses:**

* While results have been presented on class-conditional generation and frame prediction tasks, the benefit or penalty from using the proposed method on other tasks such as video editing and stylization remains unclear.
* While not being applied to videos before, the holistic token approach appears similar to [1] on images. The differences other than the AR prior part need to be clarified.

[1] Yu et al. An Image is Worth 32 Tokens for Reconstruction and Generation. arXiv 2406.07550

**Questions:**

* What are the differences between the proposed holistic tokenization approach and [1] beyond the AR prior part? Could the authors clarify from the aspects of input/output format, model architecture, training objective, etc.?
* Would the holistic tokens hurt the appearance preservation capabilities in editing tasks such as inpainting and outpainting?
* In the K600 frame prediction benchmark, what is the input to the LARP encoder at inference time? Could the authors clarify how to handle 5-frame condition with the patch size described in Sec. 4.1?

[1] Yu et al. An Image is Worth 32 Tokens for Reconstruction and Generation. arXiv 2406.07550

---

> ### Author Response · Authors · 2024-11-18
> **Official Comment by Authors**
>
> We thank Reviewer qHTB for their insightful and constructive review of our paper. We are grateful for the positive remarks on our novel approach to video tokenization, its compatibility with AR models, and the strong performance of our method. We address each of the raised concerns below.
>
> > W1: Benefit or penalty from using the proposed method on other tasks such as video editing and stylization.
>
> Following recent works on video tokenization [6,7], we primarily focus our efforts on video generation as the primary task.
> While video editing and stylization are beyond the scope of this paper, the potential benefits of the proposed methods for these tasks are promising. Since LARP is trained for alignment with AR models, including multimodal large language models (MLLMs), it can naturally enhance instruction-driven editing when integrated with MLLMs. Instruction fine-tuning on mixed text and LARP tokens using AR NLL objectives is expected to leverage LARP's AR-aligned design, likely resulting in notable improvements in editing and stylization quality. We believe these are interesting future follow-up works of our paper.
>
> > W2 & Q1: Comparison with [1].
>
> We would like to emphasize that, as noted at the end of Section 2.1 in our original paper, [1] is a concurrent work to ours.
>
> Although [1] also uses holistic tokens to represent visual signals, LARP and [1] differ significantly in several key aspects, as outlined below:
>
> 1. **Order and Prior**: [1] does not define an order for their latent tokens and does not use prior models or regularization losses to enhance the performance of their downstream generative models. In contrast, the AR prior model is a core contribution of LARP, substantially enhancing video generation quality, as demonstrated in Figure 1(c) of our original paper.
> 1. **Generative Model**: [1] uses MaskGIT [5] as their video generator, a masked language modeling (MLM) generative model. In contrast, LARP is designed for AR generative models, known for their exceptional effectiveness and scalability.
> 1. **Focus**: [1] focuses on image generation, but LARP focuses on video generation.
> 1. **De-tokenization**: [1]'s ViT decoder operates similarly to MAE [2], where multiple ***identical mask tokens*** are input to their ViTs to reconstruct the images patches. In contrast, LARP is inspired by DETR [3] and the Q-former introduced in [4], where ***distinct query tokens*** are used to extract holistic tokens and reconstruct video patches.
> 1. **Training Strategy**: [1] employs a two-stage training process. In the first stage, they train their image tokenizer with discrete codes generated by an off-the-shelf MaskGIT-VQGAN model, akin to a model distillation approach. In the second stage, they fine-tune the decoder using the VQGAN objective. In contrast, the LARP tokenizer is trained end-to-end with a VQGAN-like objective, making it simpler and eliminating the need for other pre-trained models.
> 1. **Discriminator Design**: For the GAN loss, [1] uses a ConvNet-based discriminator, while LARP employs a transformer-based discriminator, enhancing reconstruction fidelity and training stability. We observed that training LARP with a ConvNet-based discriminator is not stable and often leads to loss divergence.
>
> In fact, we began this project well before the preprint of [1] was released.
>
>
> > Q2: Would the holistic tokens hurt the appearance preservation capabilities in editing tasks such as inpainting and outpainting?
>
> While video editing tasks are beyond the scope of this paper, we can reasonably expect that the use of holistic tokens would not hinder these tasks. Frame prediction can be considered a special case of video outpainting, specifically temporal single-direction outpainting. Experimental results in Table 1 of our main paper show that LARP achieves competitive performance on the Kinetics-600 video frame prediction benchmark. Therefore, we believe that LARP is well-suited for broader inpainting and outpainting generation tasks, as well as other video editing applications.

---

> ### Author Response · Authors · 2024-11-18
> **Official Comment by Authors  (Continued)**
>
> > Q3: Frame prediction implementation details.
>
> To create the 5-frame conditioning holistic tokens for the frame prediction task, we first take the initial 5 frames from the ground truth video, then repeat the 5th frame 11 times and pad these repetitions with the initial 5 frames (using the same padding). This forms a complete 16-frame conditioning video. This conditioning video is then input to the LARP tokenizer to generate the conditioning holistic tokens.
>
> At inference time, the AR model predicts the holistic tokens for the full video, conditioned on the holistic tokens from the padded conditioning video.
>
> This clarification has been included in Section A.2 of the revised paper.
>
>
>
> [1] Yu et al. An Image is Worth 32 Tokens for Reconstruction and Generation. arXiv 2406.07550
>
> [2] He, Kaiming, et al. "Masked autoencoders are scalable vision learners." Proceedings of the IEEE/CVF conference on computer vision and pattern recognition. 2022.
>
> [3] Carion, Nicolas, et al. "End-to-end object detection with transformers." European conference on computer vision. Cham: Springer International Publishing, 2020.
>
> [4] Li, Junnan, et al. "Blip-2: Bootstrapping language-image pre-training with frozen image encoders and large language models." International conference on machine learning. PMLR, 2023.
>
> [5] Chang, Huiwen, et al. "Maskgit: Masked generative image transformer." Proceedings of the IEEE/CVF Conference on Computer Vision and Pattern Recognition. 2022.
>
> [6] Yu, Lijun, et al. Language Model Beats Diffusion-Tokenizer is key to visual generation. ICLR 2024.
>
> [7] Wang, Junke, et al. OmniTokenizer: A Joint Image-Video Tokenizer for Visual Generation. NeurIPS 2024.
>
> **We would like to thank reviewer qHTB once again for their review, which has been instrumental in strengthening our work. Should there be any further concerns, we would be glad to address them. We respectfully ask reviewer qHTB to consider a higher rating in light of our responses and revisions.**

---

> > ### Author Response · Authors · 2024-12-02
> >
> > We sincerely appreciate your thoughtful feedback and are deeply grateful for your recognition of the technical novelty and significance of our work.
> >
> > If you believe that an adjustment to your score is warranted, we would be very grateful. Additionally, please don’t hesitate to reach out with any further questions or comments; your insights are extremely valuable to us.
> >
> > Thank you once again for your time and thoughtful consideration.

---

> > > ### Comment · Reviewer_qHTB · 2024-12-03
> > >
> > > Thank you for the detailed responses to my questions. I have no further questions and am raising my score to 8.

---

> > > > ### Author Response · Authors · 2024-12-03
> > > >
> > > > We sincerely appreciate your recognition of the strengths in our paper and your thoughtful engagement during the discussion period. Your positive evaluation and recommendation for its presentation at the conference are truly encouraging.
> > > >
> > > > We are deeply grateful for your valuable feedback and guidance throughout this process!

---

### Official Review · Reviewer_KrLv · 2024-11-08

**Soundness:** 2
**Presentation:** 3
**Contribution:** 3
**Rating:** 8
**Confidence:** 4

**Summary:**

The paper introduces LARP, a novel method for tokenizing videos for autoregressive (AR) generative models. LARP employs a holistic approach to capture global video information more effectively and incorporates an AR prior model to enhance video generation quality. The method demonstrates strong performance in generating high-quality videos and holds potential for developing models capable of integrating both video and language.

**Strengths:**

1. This paper introduces a novel method for video tokenization that moves beyond traditional patchwise encoding, making it both interesting and innovative.

2. The paper provides a comprehensive set of experiments, including video reconstruction, class-conditional video generation, and video frame prediction, effectively demonstrating LARP's capabilities across different tasks.

3. LARP achieves state-of-the-art FVD scores on benchmarks like UCF101, indicating that it is a competitive approach in the field of video generation.

**Weaknesses:**

1. The overall computational cost and complexity of training LARP, especially with the AR prior model, may be a concern.
2. It remains unclear how stable the training process is over longer periods or under different training regimes.
3. The paper may not sufficiently discuss scenarios in which LARP underperforms or fails, which is essential for understanding the model’s limitations.

**Questions:**

How does the training process scale with larger datasets or more complex video content?
What are the hardware requirements for training and deploying LARP?
Are there specific types of videos or scenarios where LARP may underperform, and if so, what are they?

---

> ### Author Response · Authors · 2024-11-18
> **Official Comment by Authors**
>
> We thank Reviewer KrLv for their thoughtful and constructive review of our paper. We appreciate the positive feedback on our paper's novelty, experimental design, and performance. Below, we address each of the concerns raised.
>
> > W1: Computational cost and complexity of training LARP, especially with the AR prior model.
>
> We acknowledge that training LARP as a video tokenizer is computationally intensive. However, LARP's training efficiency is significantly higher compared to other transformer-based tokenizers, such as OmniTokenizer [1]. For example, training the LARP-L-Long tokenizer required only two days on 8 H100 GPUs, whereas OmniTokenizer required two weeks on 8 A100 GPUs, as reported in their paper. Additionally, we emphasize that the computational demand of using the AR prior model is minimal, as it is a lightweight model with only 21.7 million parameters.
>
> The table below compares the training speeds of LARP-L with and without the AR prior model, both conducted on 4 H100 GPUs. Notably, incorporating the AR prior model results in only a 1.19 iter/s reduction in speed, a trade-off well justified by the substantial improvement in generation quality enabled by the AR prior model, as demonstrated in Figure 1(c) of our main paper.
>
> |  Model          | Training Speed |
> |:----------------|:--------------:|
> | LARP-B          | 3.74 iter/s    |
> | LARP-B-no-Prior | 4.93 iter/s    |
>
>
> > W2: It remains unclear how stable the training process is over longer periods or under different training regimes.
>
> In Figure 7 of the revised paper, we present the training loss curves for LARP across different training epochs and regimes. As illustrated, all loss curves decrease smoothly and converge by the end of training, demonstrating LARP's strong training stability. Further details are provided in Section B.1 of the revised paper.
>
> > W3 & Q3: Limitations and failure cases.
>
> While LARP significantly enhances video generation quality, it still has certain limitations. Like other transformer-based video tokenizers [1], LARP performs best with fixed-resolution videos due to the constraints of positional encoding. Additionally, artifacts may appear in LARP-generated videos when the scenes are particularly complex. Fortunately, scaling up the AR generative model is expected to improve video quality and reduce these artifacts, as suggested by the scaling laws of AR models [2,3].
>
> This discussion has been added to Section D of the revised paper.
>
> > Q1: How does the training process scale with larger datasets or more complex video content?
>
>
> In this paper, we follow standard practices established in the literature, particularly recent works on video tokenization [1,4], to evaluate LARP on the UCF101 and K600 datasets.
> We acknowledge that due to limited computational resources, we are currently unable to scale LARP to larger video datasets with more complex content. However, we believe that the extensive experimental results obtained on these widely used benchmarks are sufficient to showcase LARP's superior performance in video generation.
>
>
> > Q2: What are the hardware requirements for training and deploying LARP?
>
> Training the LARP-L-Long tokenizer takes 2 days on 8 H100 GPUs with the default global batch size of 128, or 4 days on 4 H100 GPUs with a reduced global batch size of 96.
>
> During inference, the LARP-L-Long tokenizer and the 632M LARP AR model can generate up to 1.8 videos per second on a single H100 GPU. The minimum GPU memory requirement to run the entire LARP system is only 3GB when using a batch size of 1.
>
>
>
>
>
>
> [1] Wang, Junke, et al. "OmniTokenizer: A Joint Image-Video Tokenizer for Visual Generation." arXiv preprint arXiv:2406.09399 (2024).
>
> [2] Henighan, Tom, et al. "Scaling laws for autoregressive generative modeling." arXiv preprint arXiv:2010.14701 (2020).
>
> [3] Sun, Peize, et al. "Autoregressive Model Beats Diffusion: Llama for Scalable Image Generation." arXiv preprint arXiv:2406.06525 (2024).
>
> [4] Yu, Lijun, et al. Language Model Beats Diffusion-Tokenizer is key to visual generation. ICLR 2024.
>
>
> **We thank reviewer KrLv once again for their review, which has been invaluable in improving our work. If there are any additional concerns, we would be more than happy to address them. We kindly ask reviewer KrLv to consider an increased rating based on our responses and revisions.**

---

> > ### Comment · Reviewer_KrLv · 2024-11-22
> >
> > Thank you for the author's response. All of my questions have been addressed, and I would like to increase my score to 8.

---

> > > ### Author Response · Authors · 2024-11-25
> > >
> > > We sincerely thank you for your thoughtful feedback and for engaging with our responses! We appreciate your recognition of LARP’s novelty and performance, as well as your increased rating.

---

### Meta-Review · Area_Chair_F8j5 · 2024-12-17

**Metareview:**

This paper presents a method for learning tokenization that goes beyond patch-based representations and integrates a lightweight AR model, demonstrating improvements in AR generative modeling.

Strengths:

* The efficacy of the proposed method is well-validated.
* Comprehensive experiments achieve state-of-the-art results.
* The authors provide an insightful analysis of two key limitations of AR tokenization.

In the end, all reviewers are in favor of accepting this submission. The topic of the paper may attract interest in the field of visual generation using large language models.

**Additional Comments On Reviewer Discussion:**

Post-rebuttal discussions indicate that no major concerns remain. The remaining issues appear to be minor, primarily regarding the interpretation of the heatmap visualization.

---

### Decision · Program_Chairs · 2025-01-22

Accept (Oral)